# Fair Bilevel Neural Network (FairBiNN): On Balancing fairness and accuracy via Stackelberg Equilibrium

**Mehdi Yazdani-Jahromi**
Department of Computer Science
University of Central Florida
Orlando, FL 32816
yazdani@ucf.edu

**Ali Khodabandeh Yalabadi**
Department of Industrial Engineering
University of Central Florida
Orlando, FL 32816
yalabadi@ucf.edu

**AmirArsalan Rajabi**
Department of Computer Science
University of Central Florida
Orlando, FL 32816
am954283@ucf.edu

**Aida Tayebi**
Department of Industrial Engineering
University of Central Florida
Orlando, FL 32816
ai530737@ucf.edu

**Ivan Garibay**
Department of Industrial Engineering
University of Central Florida
Orlando, FL 32816
igaribay@ucf.edu

**Ozlem Ozmen Garibay**
Department of Industrial Engineering
University of Central Florida
Orlando, FL 32816
ozlem@ucf.edu

## Abstract

The persistent challenge of bias in machine learning models necessitates robust solutions to ensure parity and equal treatment across diverse groups, particularly in classification tasks. Current methods for mitigating bias often result in information loss and an inadequate balance between accuracy and fairness. To address this, we propose a novel methodology grounded in bilevel optimization principles. Our deep learning-based approach concurrently optimizes for both accuracy and fairness objectives, and under certain assumptions, achieving proven Pareto optimal solutions while mitigating bias in the trained model. Theoretical analysis indicates that the upper bound on the loss incurred by this method is less than or equal to the loss of the Lagrangian approach, which involves adding a regularization term to the loss function. We demonstrate the efficacy of our model primarily on tabular datasets such as UCI Adult and Heritage Health. When benchmarked against state-of-the-art fairness methods, our model exhibits superior performance, advancing fairness-aware machine learning solutions and bridging the accuracy-fairness gap. The implementation of FairBiNN is available on https://github.com/yazdanimehdi/FairBiNN.

## 1 Introduction

Artificial intelligence and machine learning models have seen significant growth over the past decades, leading to their integration into various domains such as hiring pipelines, face recognition, financial services, healthcare, and criminal justice. This widespread adoption of algorithmic decision-making has raised concerns about algorithmic bias, which can result in discrimination and unfairness towards minority groups. Recently, the issue of fairness in artificial intelligence has garnered considerable

38th Conference on Neural Information Processing Systems (NeurIPS 2024).

attention from interdisciplinary research communities, addressing these ethical concerns [50]. Several definitions of fairness have been proposed to tackle unwanted bias in machine learning techniques. These definitions generally fall into two categories: individual fairness and group fairness. Individual fairness ensures that similar individuals are treated similarly, with similarities determined by past information [20, 73]. Group fairness, on the other hand, measures statistical equality between different subgroups defined by sensitive characteristics such as race or gender [76, 41, 26]. In this paper, we focus on group fairness, which we will refer to simply as fairness from this point onward.

Fairness approaches in machine learning are commonly categorized into three groups: (1) Pre-process approaches: These methods involve changing the data before training to improve fairness, such as reweighing labels or adjusting features to reduce distribution differences between privileged and unprivileged groups, making it harder for classifiers to differentiate them [34, 42, 22, 63]. Generative adversarial networks were also utilized to produce unbiased datasets by altering the generator network's value function to balance accuracy and fairness [53]. (2) In-process approaches: These methods modify the algorithm during training, for instance by adding regularization terms to the objective function to ensure fairness. Examples include penalizing the mutual information between protected attributes and classifier predictions to allow a trade-off between fairness and accuracy [35], and adding constraints to satisfy a proxy for equalized odds [74, 75]. (3) Post-process approaches: These techniques adjust the outcomes after training, such as flipping some outcomes to improve fairness [26], or using different thresholds for privileged and unprivileged groups to optimize the trade-off between accuracy and fairness [45, 11].

In this work we targeted the in-process bias mitigation category. Traditionally, the fairness multi-criteria problem has been addressed using Lagrangian optimization, wherein the objective function is a weighted sum of the primary and secondary loss functions. While this approach allows for the explicit incorporation of fairness constraints through Lagrange multipliers, it may overlook the complex interdependencies between the primary and secondary objectives.
A promising alternative to the Lagrangian is the bilevel optimization approach which offers several advantages. By formulating the problem as a hierarchical optimization task, we can explicitly model the interactions between the primary and secondary objectives. This allows us to capture the nuanced dynamics of fairness optimization and ensure that improvements in one objective do not come at the expense of the other.

In summary, we introduce a novel method that can be trained on existing datasets without requiring any alterations to the data itself (data augmentation, perturbation, etc). Our methodology provides a principled approach to addressing the multi-criteria fairness problem in neural networks. Through rigorous theoretical analysis, we formulated the problem as a bilevel optimization task, proving that it yields Pareto-optimal solutions. We derived an effective optimization strategy that is at least as effective as the Lagrangian approach. Empirical evaluations on tabular datasets demonstrate the efficacy of our method, achieving superior results compared to traditional approaches.

## 2    Related works

Multi-objective optimization in neural networks involves optimizing two or more conflicting objectives simultaneously. Fairness problems are inherently multi-objective in nature, as improvements in one objective (e.g., enhancing fairness) often come at the expense of another objective (e.g., improving accuracy). Several optimization techniques in neural networks have been employed to balance accuracy and fairness. Classic Methods transform these objectives into a single objective by combining them, typically using a weighted sum where each objective is multiplied by a weight that reflects its importance. Adding Regularization and penalty terms are the most common methods that incorporate fairness constraints (e.g., demographic parity, equal opportunity) directly into the loss function, penalizing disparities in prediction errors across demographic groups or any other unfair behavior. To reduce variation across different groups, Zafar et al. [74] proposes "disparate mistreatment", a new notation for fairness, and standardized the decision bounds of a convex margin-based classifier. Adversarial debiasing and Fair Representation Learning are two examples of these techniques, which encourage the model to generate fair outcomes by introducing a penalty term based on an adversarial network or a representation learning framework that is invariant to protected attributes, respectively. Zhang et al. [77] addressed bias by limiting an adversary's ability to infer sensitive characteristics from predictions. Avoiding the complexity of adversarial training, Moyer et al. [47] used mutual information to achieve invariant data representations concerning specific factors. Song et al. [61] proposed

an information-theoretic method that leverages both information-theoretic and adversarial approaches to achieve controllable fair data representations, adhering to demographic parity. By incorporating a forget-gate similar to those in LSTMs, Jaiswal et al. [30] introduced adversarial forgetting to enhance fairness. Gupta et al. [25] utilized certain estimates for contrasting information to optimize theoretical objectives, facilitating suitable trade-offs between demographic parity and accuracy in the statistical population. Lagrangian optimization techniques are a subset of these techniques that use Lagrange multipliers or other similar techniques to incorporate constraints directly into the objective function, turning constrained optimization problems into unconstrained ones. Agarwal et al. [2] proposes an approach for fair classification by framing the constrained optimization problem as a two-player game where one player optimizes the model parameters, and the other imposes the constraints, and Lagrangian multipliers are used to solve this problem. Cotter et al. [12] expanded this work in a more general inequality-constrained setting, by simultaneously training each player on two distinct datasets to enhance generalizability. They enforce independence by regularizing the covariance between predictions and sensitive variables, which reduces the variation in the relationship between the two. Despite analytic solutions and theoretical assurances, scaling game-theoretic techniques for more complex models remains challenging [9]. In addition, these constraints-based optimizations are data-dependent, meaning the model may exhibit different behavior during evaluation even if constraints are met during training. Less common approaches including Pareto-based genetic algorithm, Reinforcement Learning, Gradient-Based Methods, and Transfer and Meta-Learning Approaches have been also utilized in this domain. Mehrabi et al. [43] demonstrated how proxy attributes lead to indirect unfairness using an attention-based approach and employed a post-processing method to reduce the weight of attributes responsible for unfairness. Perrone et al. [49] introduces a general constrained Bayesian optimization (BO) framework to fine-tune the model's performance while enforcing one or multiple fairness constraints. A probabilistic model is used to describe the objective function, and estimates are made for the posterior variances and means for each hyperparameter configuration. By adding a fairness regularization term to a meta-learning framework, Slack et al. [60] suggests an adaptation of the Model-Agnostic Meta-Learning (MAML) [23] algorithm. The primary objective and fairness regularization terms are included in the loss function used to update the model parameters for each task during the inner loop (Learner). The model parameters are updated in the outer loop (Meta-learner) to maximize performance and fairness across all tasks. Although these techniques have achieved a good balance between fairness and accuracy, they might not capture all of the complex interdependencies between these two objectives. In this paper we propose a bilevel optimization approach as an alternative to the Lagrangian approaches. Bilevel optimization is a hierarchical structure in which the context or constraints for the "follower" (lower-level) problem are set by the "leader" (upper-level) problem [17]. The leader makes decisions first, and the follower optimizes their decisions based on the leader's choices. This approach can handle more complex and nuanced multi-objective optimization problems in neural networks and is suitable for scenarios where one objective directly influences another and there are complex interactions between the two objectives. In this paper we demonstrate that the bilevel optimization often can achieve better balance and performance compared to classic regularization-based optimization approaches [17, 59, 10]. Bilevel optimization offers several advantages; by explicitly modeling a two-level decision-making process, his approach represents the problems in a more natural way where one objective inherently depends on the outcome of another. It provides more flexibility and control over the optimization process by enabling separate optimization of constraints at each level. The upper-level optimization can dynamically adjust the lower-level objective based on the current solution, potentially leading to more adaptive and context-sensitive optimization outcomes. Fairness and accuracy objectives can be directly integrated into the optimization framework without the need for additional strategies such as meta-learning.

## 3   Methodology

In this section we are introducing a novel bi-level optimization framework for training neural networks to obtain Pareto optimal solutions when optimizing two potentially competing objectives. Our approach leverages a leader-follower structure, where the leader problem aims to minimize one objective function (e.g. a primary loss), while the follower problem optimizes a secondary objective. We provide theoretical guarantees that our bi-level approach produces Pareto optimal solutions and performs at least as well as, and often strictly better than, the common practice of combining multiple objectives via a weighted regularization term in a single loss function. The full statements

of these theorems and their proofs are provided in the Theoretical Analysis subsection below. Our bi-level approach offers several benefits over regularization-based methods. First, it allows for easy customization of the architecture and training algorithm used for each objective. The leader and follower problems can utilize different network architectures, regularizers, optimizers, etc. as best suited for each task. Second, the leader problem remains a pure minimization of the primary loss, without any regularization terms that may slow or hinder its optimization. Separating out secondary objectives ensures the primary task is learned most effectively. Finally, bi-level training exposes a clear interface for controlling the trade-off between objectives. By constraining the follower problem more or less strictly, we can encourage stronger or weaker adherence to the secondary goal relative to the primary one. To realize these benefits, we employ an iterative gradient-based algorithm to solve the bi-level problem, alternating between updating the leader and follower parameters. We unroll the follower optimization for a fixed number of steps, and backpropagate through this unrolled process to update the leader weights.

## 3.1 Theoretical Analysis

### 3.1.1 Problem Formulation

The fairness multi-criteria problem in neural networks can be formulated as a bi-criteria optimization problem. Let $f(\theta_p, \theta_s)$ denote the primary objective loss function and $\varphi(\theta_p, \theta_s)$ denote the secondary objective loss function. Here, $\theta_p \in \Theta_p$ represents the parameters responsible for optimizing the primary objective, and $\theta_s \in \Theta_s$ represents the parameters for the secondary objective. The problem is formally stated as:

$$\min_{\theta_p \in \Theta_p, \theta_f \in \Theta_f} \{f(\theta_p, \theta_f), \varphi(\theta_p, \theta_f)\} \tag{1}$$

### 3.1.2 Theoretical Foundation: Stackelberg Equilibrium and Pareto Optimality

We leverage the theoretical results of [46], which investigates the relationship between Stackelberg equilibria and Pareto optimality in game theory. The paper addresses fundamental questions regarding the conditions under which a Stackelberg equilibrium coincides with a Pareto optimal outcome. By proving that our bi-level optimization problem satisfies the assumptions outlined in the paper, we establish a strong theoretical foundation for our approach. Specifically, we demonstrate that under certain conditions, the Stackelberg equilibrium of our bi-level optimization problem is equivalent to a Pareto optimal solution for the bi-criteria problem of balancing accuracy and fairness objectives. By rigorously verifying these assumptions in the context of our neural network optimization problem, we establish that the Stackelberg equilibrium reached by our bilevel approach indeed corresponds to a Pareto optimal solution. This theoretical grounding provides confidence that our methodology effectively balances the competing objectives of accuracy and fairness, yielding a principled and well-justified solution to the problem at hand.

We leverage several key theoretical results to formulate our approach. First, Lemma 3.5 establishes the Lipschitz continuity of the neural network function with respect to a subset of parameters. This lemma provides the foundation for analyzing the behavior of the objective functions under parameter variations.

**Definition 3.1.** A function $f : \mathbb{R}^n \longrightarrow \mathbb{R}^m$ is called Lipschitz continuous if there exists a constant L such that:

$$\forall x, y \in \mathbb{R}^n, \|f(x) - f(y)\|_2 \leq L \|x - y\| \tag{2}$$

The smallest $L$ for which the previous inequality is true is called the Lipschitz constant of $f$ and will be denoted $L(f)$.

Assume that the following assumptions are satisfied:

**Assumption 3.2.** The primary loss function $f(\theta_p, \theta_s)(x)$ is strictly convex in a neighborhood of its local optimum. That is, for any $\theta_p, \theta_p' \in \Theta_p$ and fixed $\theta_s \in \Theta_s$, if $\theta_p \neq \theta_p'$ and $\theta_p, \theta_p'$ are sufficiently close to the local optimum $\theta_p^*$, then

$$f(\lambda\theta_p + (1 - \lambda)\theta_p', \theta_s) < \lambda f(\theta_p, \theta_s) + (1 - \lambda)f(\theta_p', \theta_s) \tag{3}$$

for any $\lambda \in (0, 1)$.

**Assumption 3.3.** $|\theta_s - \hat{\theta}_s| \leq \epsilon$, where $\epsilon$ is sufficiently small, i.e., the steps of the secondary parameters are sufficiently small. $\theta_s$ and $\hat{\theta}_s$ represent the parameters for the secondary objective and their updated values, respectively.

**Assumption 3.4.** Let $f_l(.)$ denote the output function of the $l$-th layer in a neural network with $L$ layers. For each layer $l \in 1, \ldots, L$, there exists a constant $c_l > 0$ such that for any input $x_l$ to the $l$-th layer:

$$|f_l(x_l)| \leq c_l \tag{4}$$

where $|.|$ denotes a suitable norm (e.g., Euclidean norm for vectors, spectral norm for matrices). Refer to Section A.4 for common practices in implementing the bounded output assumption.

We recognized the importance of examining how our theory's underlying assumptions apply to real-world applications. For a detailed discussion, refer to Section A.3.

**Lemma 3.5.** *Let $f(x; \theta)$ be a neural network with L layers, where each layer is a linear transformation followed by a Lipschitz continuous activation function.*
*Let $\theta$ be the set of all parameters of the neural network, and $\theta_s \subseteq \theta$ be any subset of parameters. Then, $f(x; \theta)$ is Lipschitz continuous with respect to $\theta_s$. [See proof A.4]*

We discussed the Lipschitz continuity of common activation functions and popular neural networks, such as CNNs and GNNs, in Sections A.5 and A.6, respectively.

Theorems 3.6 and 3.7 further inform our approach. The former establishes conditions under which improvements in the secondary objective lead to improvements in the primary objective, while the latter guarantees the existence of unique minimum solutions for the secondary loss function under certain optimization conditions.

**Theorem 3.6.** *Let $f(\theta_p, \theta_s)$ for constant $\theta_s$ be the primary objective loss function and $\varphi(\theta_p, \theta_s)$ for constant $\theta_p$ be the secondary objective loss function, where $\theta_p \in \Theta_p$ and $\theta_s \in \Theta_s$ are the primary task and secondary task parameters, respectively.*

*Consider two sets of parameters $(\theta_p, \theta_s)$ and $(\hat{\theta}_p, \hat{\theta}_s)$ such that $\varphi(\hat{\theta}_p, \hat{\theta}_s) \leq \varphi(\theta_p, \theta_s)$. Then $f(\hat{\theta}_p, \hat{\theta}_s) \leq f(\theta_p, \theta_s)$ holds based on Lemma 3.5. [See proof A.5]*

**Theorem 3.7.** *Let $\varphi(\theta_p, \theta_s)$ be the secondary loss function, where $\theta_p \in \Theta_p$ and $\theta_s \in \Theta_s$ are the primary and secondary task parameters, respectively. Let $(\theta_p^{(t)}, \theta_s^{(t)})$ denote the parameters at optimization step t, and let $(\theta_p^{(t+1)}, \theta_s^{(t+1)})$ be the updated parameters obtained by minimizing $\varphi(\theta_p^{(t)}, \theta_s)$ with respect to $\theta_s$ using a sufficiently small step size $\eta > 0$, i.e.:*

$$\theta_s^{(t+1)} = \theta_s^{(t)} - \eta \nabla_{\theta_s} \varphi(\theta_p^{(t)}, \theta_s^{(t)}) \tag{5}$$

*Then, for a sufficiently small step size $\eta$, the updated secondary parameters $\theta_s^{(t+1)}$ are the unique minimum solution for the secondary loss function $\varphi(\theta_p^{(t)}, \theta_s)$. [See proof A.6]*

Based on these theoretical insights, we derive our bilevel optimization formulation, as described in Theorem 3.8. This theorem establishes the equivalence between the bi-criteria problem and a bilevel optimization problem, allowing us to apply existing theoretical results on Stackelberg equilibrium to the optimization of neural networks.

**Theorem 3.8.** *Under the assumptions stated in Theorems 3.6 and 3.7, the bi-criteria problem (Eq. (1)) is equivalent to the bilevel optimization problem:*

$$\min_{\theta_p \in \Theta_p} \quad f(\theta_p, \theta_s^*(\theta_p)) \tag{6}$$

$$s.t. \quad \theta_s^*(\theta_p) = \arg\min_{\theta_s \in \Theta_s} \varphi(\theta_p, \theta_s) \tag{7}$$

*where $\theta_s^*(\theta_p)$ denotes the optimal secondary parameters for a given $\theta_p$.*

*Proof.* The proof follows from Theorems 3.6 and 3.7 [46].

By Theorem 3.6, under the assumptions of strict convexity, Lipschitz continuity, and sufficiently small steps of the secondary parameters, if $\varphi(\hat{\theta}_p, \hat{\theta}_s) \leq \varphi(\theta_p, \theta_s)$, then $f(\hat{\theta}_p, \hat{\theta}_s) \leq f(\theta_p, \theta_s)$.

By Theorem 3.7, under the same assumptions, for each optimization step of the secondary loss function with sufficiently small steps, the updated parameters are the unique minimum solution for the secondary loss function, then the bi-criteria problem (1) is equivalent to the bilevel optimization problem.

Therefore, the conclusions drawn in the paper [46] can be directly applied to the multi-objective optimization problem in neural networks, as the problem is equivalent to the bilevel optimization problem under the stated assumptions. □

**Theorem 3.9.** *Assume that the step size in the Lagrangian approach $\alpha_{\mathcal{L}}$ is equal to the step size for the outer optimization problem in the bilevel optimization approach $\alpha_f$, the scale of the two loss functions should be comparable, the Lagrangian multiplier $\lambda$ is equal to the step size for the inner optimization problem in the bilevel optimization approach $\alpha_s$, and $\theta_p$ is overparameterized for the given problem. Then, under certain conditions, the overall performance of the primary loss function in the bilevel optimization approach may be better than the Lagrangian approach.*

*Proof.* Let $f(\theta_p, \theta_s)$ denote the primary loss and $\varphi(\theta_p, \theta_s)$ denote the secondary loss. Assume that both $f$ and $\varphi$ are differentiable with respect to $\theta_p$ and $\theta_s$. Define the Lagrangian function as:

$$\mathcal{L}(\theta_p, \theta_s, \lambda) = f(\theta_p, \theta_s) + \lambda \varphi(\theta_p, \theta_s) \tag{8}$$

The update rules for $\theta_p$ and $\theta_s$ in the Lagrangian approach are:

$$\theta_p^{(t+1)} = \theta_p^{(t)} - \alpha_{\mathcal{L}} \nabla_{\theta_p} \mathcal{L}(\theta_p^{(t)}, \theta_s^{(t)}, \lambda) \tag{9}$$

$$\theta_s^{(t+1)} = \theta_s^{(t)} - \alpha_{\mathcal{L}} \nabla_{\theta_s} \mathcal{L}(\theta_p^{(t)}, \theta_s^{(t)}, \lambda) \tag{10}$$

The update rules for $\theta_p$ and $\theta_s$ in the bilevel optimization approach are:

$$\theta_p^{(t+1)} = \theta_p^{(t)} - \alpha_f \nabla_{\theta_p} f(\theta_p^{(t)}, \theta_s^{(t)}) \tag{11}$$

$$\theta_s^{(t+1)} = \theta_s^{(t)} - \alpha_s \nabla_{\theta_s} \varphi(\theta_p^{(t)}, \theta_s^{(t)}) \tag{12}$$

Due to the overparameterization of $\theta_p$, there exists a set $\Theta_p$ such that for any $\theta_p \in \Theta_p$ [3], $f(\theta_p, \theta_s) = f(\theta_p^*, \theta_s)$, where $\theta_p^*$ is an optimal solution for the primary loss when $\theta_s$ is fixed. Suppose that the bilevel optimization approach converges to a solution $(\theta_p^B, \theta_s^B)$ and the Lagrangian approach converges to a solution $(\theta_p^L, \theta_s^L)$. Consider the following inequality:

$$f(\theta_p^B, \theta_s^B) = f(\theta_p^B, \theta_s^B) + \lambda \varphi(\theta_p^B, \theta_s^B) - \lambda \varphi(\theta_p^B, \theta_s^B) \tag{13}$$

$$\leq f(\theta_p^B, \theta_s^B) + \lambda \varphi(\theta_p^B, \theta_s^B) - \lambda \varphi(\theta_p^L, \theta_s^B) \tag{14}$$

$$= \mathcal{L}(\theta_p^B, \theta_s^B, \lambda) - \lambda \varphi(\theta_p^L, \theta_s^L) \tag{15}$$

$$\leq \mathcal{L}(\theta_p^L, \theta_s^L, \lambda) - \lambda \varphi(\theta_p^L, \theta_s^L) \tag{16}$$

$$= f(\theta_p^L, \theta_s^L) \tag{17}$$

The first inequality holds because $(\theta_p^L, \theta_s^L)$ is the minimizer of $\varphi(\theta_p, \theta_s)$ in the Lagrangian approach. The second inequality holds because $(\theta_p^L, \theta_s^L)$ is the minimizer of $\mathcal{L}(\theta_p, \theta_s, \lambda)$ in the Lagrangian approach. Since $\theta_p^B \in \Theta_p$ and $\theta_p^L \notin \Theta_p$, we have:

$$f(\theta_p^B, \theta_s^B) = f(\theta_p^B, \theta_s^B) \leq f(\theta_p^L, \theta_s^L) \tag{18}$$

Therefore, under the assumptions that $\alpha_{\mathcal{L}} = \alpha_f$, The sizes of the two loss functions $f(\theta_p, \theta_s)$ and $\varphi(\theta_p, \theta_s)$ should not differ significantly in terms of their order of magnitude, $\lambda = \alpha_s$, and $\theta_p$ is overparameterized for the given problem, the bilevel optimization approach may converge to a solution that achieves better performance for the primary loss compared to the Lagrangian approach. □

## 3.2 Practical Implementation

To connect the Stackelberg game analysis with a practical implementation for datasets, we can formulate a bilevel optimization problem. The upper-level problem corresponds to the accuracy player (leader), while the lower-level problem corresponds to the fairness player (follower). We'll use gradient-based optimization techniques to solve the problem.

Let's consider a dataset $\mathcal{D} = \{(x_i, a_i, y_i)\}_{i=1}^N$, where $x_i$ represents the features, $a_i$ represents the sensitive attribute, and $y_i$ represents the target variable for the $i$-th sample.

The optimization problem can be formulated as follows:

$$\min_{\theta_a} \quad \frac{1}{N} \sum_{i=1}^N L_{acc}(f(x_i; \theta_a, \theta_f^*), y_i) \tag{19}$$

$$\text{s.t.} \quad \theta_f^* \in \arg\min_{\theta_f} \frac{1}{N} \sum_{i=1}^N L_{fair}(f(x_i; \theta_a, \theta_f), a_i, y_i) \tag{20}$$

where $f(x; \theta_a, \theta_f)$ is the model parameterized by the accuracy parameters $\theta_a$ and fairness parameters $\theta_f$, $L_{acc}$ is the accuracy loss function (e.g., binary cross-entropy), and $L_{fair}$ is the fairness loss function (e.g., demographic parity loss). We showed that the demographic parity loss function, when applied to the output of neural network layers, is also Lipschitz continuous (Theorem 3.10).

**Demographic Parity Loss Function:** The demographic parity loss function $DP(f)$ is defined as:

$$DP(f) = \left| \mathbb{E}_{x \sim p(x|a=0)}[f(\theta_1; x)] - \mathbb{E}_{x \sim p(x|a=1)}[f(\theta_2; x)] \right| \tag{21}$$

where $a$ is a sensitive attribute (e.g., race, gender) with two possible values (0 and 1), and $p(x|a)$ is the conditional probability distribution of $x$ given $a$.

**Theorem 3.10.** *If $f(x)$ is Lipschitz continuous with Lipschitz constant $L_f$, then the demographic parity loss function $\ell_{DP}(f)$ is also Lipschitz continuous with Lipschitz constant $L_{DP} = 2L_f$. [See proof A.9]*

*We can easily extend this theorem to include another common fairness metric, equalized odds, as explained in Section A.2.*

Here's a practical implementation using gradient-based optimization:

---
**Algorithm 1** Fairness-Accuracy Bilevel Optimization
---
1: Initialize accuracy parameters $\theta_a$ and fairness parameters $\theta_f$
2: **while** not converged or max iterations $T$ not reached **do**
3:     Accuracy player's optimization
4:     Sample a minibatch $\mathcal{B}_a \subset \mathcal{D}$
5:     Compute accuracy loss: $L_a = \frac{1}{|\mathcal{B}_a|} \sum_{i \in \mathcal{B}_a} L_{acc}(f(x_i; \theta_a, \theta_f), y_i)$
6:     Update accuracy parameters: $\theta_a \leftarrow \theta_a - \eta_a \nabla_{\theta_a} L_a$
7:     Fairness player's optimization
8:     Sample a minibatch $\mathcal{B}_f \subset \mathcal{D}$
9:     Compute fairness loss: $L_f = \frac{1}{|\mathcal{B}_f|} \sum_{i \in \mathcal{B}_f} L_{fair}(f(x_i; \theta_a, \theta_f), a_i, y_i)$
10:    Update fairness parameters: $\theta_f \leftarrow \theta_f - \eta_f \nabla_{\theta_f} L_f$
11: **end while**
---

In practice, the model $f(x; \theta_a, \theta_f)$ can be implemented as a neural network with separate layers for accuracy and fairness (Figure 8). The accuracy layers are parameterized by $\theta_a$, while the fairness layers are parameterized by $\theta_f$. The accuracy loss $L_{acc}$ can be chosen based on the task at hand, such as binary cross-entropy for binary classification or mean squared error for regression. The fairness loss $L_{fair}$ can be a fairness metric such as demographic parity loss or equalized odds loss. The learning rates $\eta_a$ and $\eta_f$ control the step sizes for updating the accuracy and fairness parameters, respectively. They can be tuned using techniques like grid search or learning rate scheduling.

By implementing this algorithm on a dataset, we can optimize the model to balance accuracy and fairness, guided by the Stackelberg game formulation. At each iteration, the parameters related to accuracy are optimized while keeping the fairness parameters fixed. Then, with the accuracy parameters held constant, the fairness parameters are optimized. This separate optimization process provides fine-grained control over the trade-off between accuracy and fairness.

## 4   Experiments

In this section, we contrast our methodology with other benchmark approaches found in the literature.

We employed two metrics for evaluation: accuracy (higher values preferred) for the classification task, and demographic parity differences (DP, lower values preferred) for fairness assessment. Detailed descriptions of all metrics used and implementation settings are available in sections A.9 and A.10 in the appendix, respectively.

We evaluated our method for bias mitigation to various current state-of-the-art approaches. We concentrate on strategies specifically tuned to achieve the best results in statistical parity metrics on tabular studies.

### 4.1   Datasets:

We used two well-known benchmark datasets in this field for our experiments which are as follows: *UCI Adult Dataset* [19], This dataset is based on demographic data gathered in 1994, including a train set of 30000 and a test set of 15,000 samples. The goal is to forecast if the salary is more than $50,000 yearly, and the binary protected attribute is the gender of samples gathered in the dataset.

*Heritage Health Dataset* [1], Predicting the Charleson Index, a measure of a patient's 10-year mortality. The Heritage Health dataset contains samples from roughly 51,000 patients of which 41000 are in the training set, and 11000 are in the test set. The protected attribute, which has nine potential values, is age.

### 4.2   Baselines:

We compare our results with the following state-of-the-art methods as benchmarks:

- **CVIB** [47]: Achieves fairness using a conditional variational autoencoder.
- **MIFR** [61]: Optimizes the fairness objective with a mix of information bottleneck factor and adversarial learning.
- **FCRL** [25]: Uses specific approximations for contrastive information to maximize theoretical goals, facilitating appropriate trade-offs among statistical parity, demographic parity, and precision.
- **MaxEnt-ARL** [56]: Employs adversarial learning to mitigate unfairness.
- **Adversarial Forgetting** [30]: Uses adversarial learning techniques for fairness.
- **Fair Consistency Regularization (FCR)** [4]: Aims to minimize and balance consistency loss across groups.
- **Robust Fairness Regularization (RFR)** [31]: Considers the worst-case scenario within the model weight perturbation ball for each sensitive attribute group to ensure robust fairness.

### 4.3   Bilevel (FairBiNN) vs. Lagrangian Method

We compare our proposed FairBiNN method with the traditional Lagrangian regularization approach to empirically validate the theoretical benefits of bilevel optimization. Our analysis focuses on the convergence behavior and stability of both methods. For comprehensive details on performance and computational complexity comparison, refer to Section A.7. In the appendix, we have provided the BCE loss plots over epochs for each dataset (Fig. 2) and demonstrated the superior performance of the Bi-level approach compared to the Lagrangian approach. We have also presented a comparative analysis of these approaches for the trade-off between accuracy and Statistical Parity Difference (SPD) in Figure 3.

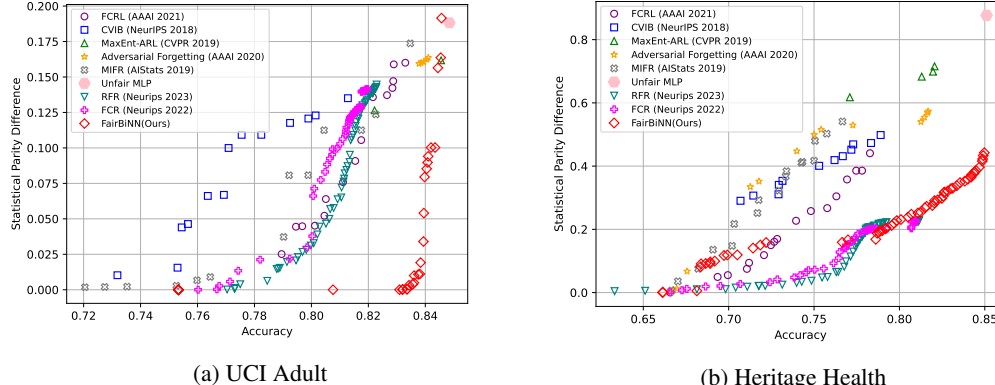

|  | (a) UCI Adult | (b) Heritage Health |
|--|--|--|

Figure 1: Accuracy of various benchmark models compared to the FairBiNN model versus statistical demographic parity for the (a) UCI Adult dataset and (b) Heritage Health dataset. The optimal region on this graph is the bottom right, indicating high accuracy and low DP. The results demonstrate that our model (red diamond markers) significantly outperforms other benchmark models on the UCI Adult dataset and closely competes with recent state-of-the-art models on the Heritage Health dataset.

While Theorem 3.9 in the paper proves that, under certain assumptions, the primary loss function in the bilevel optimization approach is upper bounded by the loss of the Lagrangian approach at the optimal solution, it does not analyze or guarantee the convergence behavior of the algorithms. The empirical results for the Health and Adult datasets show that the bilevel approach outperforms the Lagrangian method in minimizing the BCE loss. However, further investigation is needed to understand the convergence properties of the algorithms and connect the theoretical results with empirical observations. Despite this, the experimental results highlight the potential of the bilevel optimization framework to optimize accuracy and fairness objectives, offering a promising approach to address the multi-criteria fairness problem in neural networks.

### 4.3.1 Benchmark Comparison

We provide average accuracy as a measure of most probable accuracy and maximum demographic parity as a measure of worst-case scenario bias, calculated across five iterations of the training process using random seeds. Unlike Gupta et al. [25], we did not use any preprocessing on the data before feeding it to our network. Reported results for our model are Pareto solutions for the neural network during training with different $\eta_f$. Results are reported for methods with a multi-layer perceptron classifier with two hidden layers.

Figures 1a and 1b show trade-offs of the statistical demographic parity vs. accuracy associated with various bias reduction strategies in the UCI Adult dataset and Heritage Health dataset, respectively. The ideal area of the graph for the result of a method is to measure how much the curve is located in the lower right corner of the graph, which means accurate and fair results concerning protected attributes. Our results demonstrate that the Bilevel design significantly outperforms competing methods in Adult dataset.

## 5 Limitations and Future Work

While our results are promising, it's important to acknowledge several limitations of our current approach:

One of the most widely used activation functions, softmax, is not Lipschitz continuous. This limits the direct application of our method to multiclass classification problems. Future work could explore alternative activation functions or modifications to the softmax that preserve Lipschitz continuity while maintaining similar functionality for multiclass problems.

Attention mechanisms, which are widely used in modern language models and other architectures, are not Lipschitz continuous. This presents a challenge for extending our method to state-of-the-art architectures in natural language processing and other domains that heavily rely on attention. However, research into the Lipschitz continuity of attention layers has already begun, with Dasoulas et al. [16] introducing LipschitzNorm, a simple and parameter-free normalization technique applied to attention scores to enforce Lipschitz continuity in self-attention mechanisms. Their experiments on graph attention networks (GAT) demonstrate that enforcing Lipschitz continuity generally enhances the performance of deep attention models.

Our theoretical analysis primarily provides guarantees in comparison to regularization methods. While the results show improvements in fairness overall, the theory does not offer absolute fairness guarantees for the final model. Extending the theoretical framework to include direct fairness guarantees could strengthen the method's applicability.

This method was not validated on dataset augmentation approaches, which are common in practice for improving model generalization and robustness. Future work should investigate how our method interacts with various data augmentation techniques and whether it maintains its fairness properties under such conditions.

Our current implementation focuses on a single fairness metric (demographic parity). In practice, multiple, sometimes conflicting, fairness criteria may be relevant. Extending our method to handle multiple fairness constraints simultaneously could make it more versatile for real-world applications.

Addressing these limitations presents exciting opportunities for future research. By tackling these challenges, we can further enhance the applicability and effectiveness of fair machine learning methods across a broader range of real-world scenarios and cutting-edge architectures.

## 6  Discussion and Conclusion

Our primary contribution lies in the theoretical foundation and general applicability of the proposed framework, rather than extensive ablation studies on specific datasets or network configurations. However, we recognize the importance of empirical evaluations. Our work introduces a novel approach to addressing the multi-criteria fairness problem in neural networks, supported by theoretical analysis, particularly Theorem 3.8, which establishes properties of the optimal solution under certain assumptions, independent of specific datasets or architectures. The results on vision and graph datasets (A.11) and ablation studies on the impact of $\eta$ (A.13.1), the position of fairness layers (A.13.3), and different layer types (A.13), presented in the appendix, demonstrate the effectiveness and versatility of our approach. These studies show that the bilevel optimization framework can be successfully applied to various layer types and network architectures, beyond the single linear layer used in the main experiments. Our experimentation across diverse datasets, including UCI Adult, Heritage Health, and other domains like graph datasets [A.8.1] (POKEC-Z, POKEC-N, and NBA) and vision datasets [A.8.2] (CelebA), further illustrates the versatility and efficacy of our method. Including these ablation studies in the appendix allows us to maintain the main text's focus on theoretical contributions and the general framework while providing additional empirical evidence to support our claims.

Our results demonstrate the superiority of our model over state-of-the-art fairness methods in reducing bias while maintaining accuracy, highlighting the potential of our framework to advance fairness-aware machine learning solutions. Notably, our study represents a significant contribution by formulating multi-objective problems in neural networks as a bilevel design, providing a powerful tool for achieving equitable outcomes across diverse groups in classification tasks. Future research can address our method's limitations and explore potential directions as outlined in Section 5.

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

# A  Appendix / supplemental material

## A.1  Theoretical proofs

Assume that the following assumptions are satisfied:

**Assumption A.1.** The primary loss function $f(\theta_p, \theta_s)(x)$ is strictly convex in a neighborhood of its local optimum. That is, for any $\theta_p, \theta'_p \in \Theta_p$ and fixed $\theta_s \in \Theta_s$, if $\theta_p \neq \theta'_p$ and $\theta_p, \theta'_p$ are sufficiently close to the local optimum $\theta^*_p$, then

$$f(\lambda\theta_p + (1-\lambda)\theta'_p, \theta_s) < \lambda f(\theta_p, \theta_s) + (1-\lambda)f(\theta'_p, \theta_s) \tag{22}$$

for any $\lambda \in (0,1)$.

**Assumption A.2.** $|\theta_s - \hat{\theta}_s| \leq \epsilon$, where $\epsilon$ is sufficiently small, i.e., the steps of the secondary parameters are sufficiently small. $\theta_s$ and $\hat{\theta}_s$ represent the parameters for the secondary objective and their updated values, respectively.

**Assumption A.3.** Let $f_l(.)$ denote the output function of the $l$-th layer in a neural network with $L$ layers. For each layer $l \in 1, \dots, L$, there exists a constant $c_l > 0$ such that for any input $x_l$ to the $l$-th layer:

$$|f_l(x_l)| \leq c_l \tag{23}$$

where $|.|$ denotes a suitable norm (e.g., Euclidean norm for vectors, spectral norm for matrices).

**Lemma A.4.** *Let $f(x; \theta)$ be a neural network with L layers, where each layer is a linear transformation followed by a Lipschitz continuous activation function.*
*Let $\theta$ be the set of all parameters of the neural network, and $\theta_s \subseteq \theta$ be any subset of parameters. Then, $f(x; \theta)$ is Lipschitz continuous with respect to $\theta_s$.*

*Proof.* Since each activation layer is Lipschitz continuous with Lipschitz constant $L_l$ we have:

$$|f_l(x; \theta_l) - f_l(x; \theta'_l)| \tag{24}$$

$$\leq L_l|(w_l x + b_l) - (w'_l x + b'_l)| \tag{25}$$

$$= L_l|(w_l - w'_l)f_{l-1}(x) + (b_l - b'_l)| \tag{26}$$

by the triangle inequality and 24 we have:

$$L_l|(w_l - w'_l)f_{l-1}(x) + (b_l - b'_l)| \leq L_l(|w_l - w'_l||f_{l-1}(x)| + |b_l - b'_l|) \tag{27}$$

by assumption A.3 and Eq. 27 we have:

$$|f_l(x; \theta_l) - f_l(x; \theta'_l)| \leq L_l(|w_l - w'_l|c_l + |b_l - b'_l|) \leq L_l c_l |\theta_l - \theta'_l| \tag{28}$$

We can write the neural network as the composition of functions of each layer:

$$f(x; \theta) = f_l \circ f_{l-1} \circ \dots \circ f_1(x; \theta) \tag{29}$$

According to triangle inequality, we can write:

$$|f(x; \theta_i) - f(x; \theta'_i)| \leq \sum_{i=1}^{L} |f_L \circ \dots f_{i+1} \circ f_i(x; \theta_i) - f_L \circ \dots f_{i+1} \circ f_i(x; \theta'_i)| \tag{30}$$

Since the composition of Lipschitz Continuous functions is Lipschitz continuous with Lipschitz constant equal to the product of individual Lipschitz constants [24] we can write:

$$|f_L \circ \dots f_{i+1} \circ f_i(x; \theta_i) - f_L \circ \dots f_{i+1} \circ f_i(x; \theta'_i)| \leq \left( \prod_{K=i+1}^{L} L_k \right) L_i c_i |\theta_i - \theta'_i| \tag{31}$$

by using Eq. 30 and 31 we can write:

$$|f(x; \theta) - f(x; \theta')| \leq \sum_{i=1}^{L} L_i c_i \left( \prod_{k=i+1}^{L} L_k c_k \right) |\theta_i - \theta'_i| \tag{32}$$

with Cauchy–Schwarz inequality we have:

$$\sum_{i=1}^{L} L_i c_i \left( \prod_{k=i+1}^{L} L_k c_k \right) |\theta_i - \theta_i'| \tag{33}$$

$$\leq \sqrt{\sum_{i=1}^{L} (\theta_i - \theta_i')^2} \sqrt{\sum_{i=1}^{L} \left( \prod_{j=i+1}^{L} L_j \right)^2 L_i^2 c_i^2} \tag{34}$$

$$\leq L^* |\theta - \theta'| \tag{35}$$

for all $\theta = (\theta_s, \theta_{\bar{s}})$ and constant $\theta_{\bar{s}}$ we have:

$$|f(x; \theta_s) - f(x; \theta_s')| \leq L^* |\theta_s - \theta_s'| \tag{36}$$

$\square$

**Theorem A.5.** *Let $f(\theta_p, \theta_s)$ for constant $\theta_s$ be the primary objective loss function and $\varphi(\theta_p, \theta_s)$ for constant $\theta_p$ be the secondary objective loss function, where $\theta_p \in \Theta_p$ and $\theta_s \in \Theta_s$ are the primary task and secondary task parameters, respectively.*

*Consider two sets of parameters $(\theta_p, \theta_s)$ and $(\hat{\theta}_p, \hat{\theta}_s)$ such that $\varphi(\hat{\theta}_p, \hat{\theta}_s) \leq \varphi(\theta_p, \theta_s)$. Then $f(\hat{\theta}_p, \hat{\theta}_s) \leq f(\theta_p, \theta_s)$ holds based on Lemma A.4.*

*Proof.* Let $(\theta_p, \theta_s)$ and $(\hat{\theta}_p, \hat{\theta}_s)$ be two sets of parameters such that $\varphi(\hat{\theta}_p, \hat{\theta}_s) \leq \varphi(\theta_p, \theta_s)$.
By Lemma A.4, $f(\theta_p, \theta_s)$ is Lipschitz continuous with respect to $\theta_p$ and $\theta_s$.
By Assumption A.2, $|\theta_s - \hat{\theta}_s| \leq \epsilon$, where $\epsilon$ is sufficiently small. Therefore, applying the Lipschitz continuity:

$$|f(\hat{\theta}_p, \hat{\theta}_s) - f(\hat{\theta}_p, \theta_s)| \leq L|\hat{\theta}_s - \theta_s| \leq L\epsilon \tag{37}$$

Now, consider the primary loss function $f(\theta_p, \theta_s)$ for a fixed $\theta_s$. By Assumption A.1, $f(.)$ is strictly convex in a neighborhood of its local optimum $\theta_p^*$. This means that for $\hat{\theta}_p$ and $\theta_p$ sufficiently close to $\theta_p^*$:

$$f(\lambda \hat{\theta}_p + (1 - \lambda)\theta_p, \theta_s) < \lambda f(\hat{\theta}_p, \theta_s) + (1 - \lambda)f(\theta_p, \theta_s) \tag{38}$$

for any $\lambda \in (0, 1)$.

Since $\varphi(\hat{\theta}_p, \hat{\theta}_s) \leq \varphi(\theta_p, \theta_s)$, and the secondary loss function $\varphi(.)$ is used to update the parameters, we can assume that $\hat{\theta}_p$ is closer to the local optimum $\theta_p^*$ than $\theta_p$. Therefore, by the strict convexity of $f(.)$:

$$f(\hat{\theta}_p, \theta_s) \leq f(\theta_p, \theta_s) \tag{39}$$

Combining the results from (37) and (39):

$$\begin{aligned} f(\hat{\theta}_p, \hat{\theta}_s) &\leq |f(\hat{\theta}_p, \hat{\theta}_s) - f(\hat{\theta}_p, \theta_s)| + f(\hat{\theta}_p, \theta_s) \\ &\leq L\epsilon + f(\theta_p, \theta_s) \end{aligned} \tag{40}$$

As $\epsilon$ is sufficiently small, we can conclude that $f(\hat{\theta}_p, \hat{\theta}_s) \leq f(\theta_p, \theta_s)$.

Therefore, under the given assumptions, if $\varphi(\hat{\theta}_p, \hat{\theta}_s) \leq \varphi(\theta_p, \theta_s)$, then $f(\hat{\theta}_p, \hat{\theta}_s) \leq f(\theta_p, \theta_s)$. $\square$

**Theorem A.6.** *Let $\varphi(\theta_p, \theta_s)$ be the secondary loss function, where $\theta_p \in \Theta_p$ and $\theta_s \in \Theta_s$ are the primary and secondary task parameters, respectively. Let $(\theta_p^{(t)}, \theta_s^{(t)})$ denote the parameters at optimization step $t$, and let $(\theta_p^{(t+1)}, \theta_s^{(t+1)})$ be the updated parameters obtained by minimizing $\varphi(\theta_p^{(t)}, \theta_s)$ with respect to $\theta_s$ using a sufficiently small step size $\eta > 0$, i.e.:*

$$\theta_s^{(t+1)} = \theta_s^{(t)} - \eta \nabla_{\theta_s} \varphi(\theta_p^{(t)}, \theta_s^{(t)}) \tag{41}$$

*Then, for a sufficiently small step size $\eta$, the updated secondary parameters $\theta_s^{(t+1)}$ are the unique minimum solution for the secondary loss function $\varphi(\theta_p^{(t)}, \theta_s)$.*

**Assumption A.7.** $\varphi(\theta_p, \theta_s)$ is smooth and Lipschitz continuous.

*Proof.* Let $\theta_p^{(t)}$ be fixed at optimization step $t$. We consider the optimization problem of minimizing the secondary loss function $\varphi(\theta_p^{(t)}, \theta_s)$ with respect to $\theta_s$.

By the assumption, $\varphi(\theta_p^{(t)}, \theta_s)$ is smooth and Lipschitz continuous with respect to $\theta_s$. This implies that $\varphi(\theta_p^{(t)}, \theta_s)$ is continuously differentiable and its gradient $\nabla_{\theta_s} \varphi(\theta_p^{(t)}, \theta_s)$ is Lipschitz continuous with some Lipschitz constant $L > 0$, i.e., for any $\theta_s, \theta_s' \in \Theta_s$:

$$\|\nabla_{\theta_s} \varphi(\theta_p^{(t)}, \theta_s) - \nabla_{\theta_s} \varphi(\theta_p^{(t)}, \theta_s')\| \leq L\|\theta_s - \theta_s'\| \tag{42}$$

Now, consider the update rule for $\theta_s$ with a sufficiently small step size $\eta > 0$:

$$\theta_s^{(t+1)} = \theta_s^{(t)} - \eta \nabla_{\theta_s} \varphi(\theta_p^{(t)}, \theta_s^{(t)}) \tag{43}$$

We want to show that for a sufficiently small $\eta$, $\theta_s^{(t+1)}$ is the unique minimizer of $\varphi(\theta_p^{(t)}, \theta_s)$.

By the Lipschitz continuity of $\nabla_{\theta_s} \varphi(\theta_p^{(t)}, \theta_s)$ and the update rule, we have:

$$\varphi(\theta_p^{(t)}, \theta_s^{(t+1)}) \leq \varphi(\theta_p^{(t)}, \theta_s^{(t)}) + \langle \nabla_{\theta_s} \varphi(\theta_p^{(t)}, \theta_s^{(t)}), \theta_s^{(t+1)} - \theta_s^{(t)} \rangle + \frac{L}{2}\|\theta_s^{(t+1)} - \theta_s^{(t)}\|^2 \tag{44}$$

$$= \varphi(\theta_p^{(t)}, \theta_s^{(t)}) - \eta\|\nabla_{\theta_s} \varphi(\theta_p^{(t)}, \theta_s^{(t)})\|^2 + \frac{L\eta^2}{2}\|\nabla_{\theta_s} \varphi(\theta_p^{(t)}, \theta_s^{(t)})\|^2 \tag{45}$$

$$= \varphi(\theta_p^{(t)}, \theta_s^{(t)}) - \eta\left(1 - \frac{L\eta}{2}\right)\|\nabla_{\theta_s} \varphi(\theta_p^{(t)}, \theta_s^{(t)})\|^2 \tag{46}$$

If we choose $\eta < \frac{2}{L}$, then $\left(1 - \frac{L\eta}{2}\right) > 0$, and we have:

$$\varphi(\theta_p^{(t)}, \theta_s^{(t+1)}) < \varphi(\theta_p^{(t)}, \theta_s^{(t)}) \tag{47}$$

This implies that $\theta_s^{(t+1)}$ is a strict minimizer of $\varphi(\theta_p^{(t)}, \theta_s)$.

To show that $\theta_s^{(t+1)}$ is the unique minimizer, suppose there exists another minimizer $\tilde{\theta}_s \neq \theta_s^{(t+1)}$. By the strict inequality above, we must have:

$$\varphi(\theta_p^{(t)}, \tilde{\theta}_s) > \varphi(\theta_p^{(t)}, \theta_s^{(t+1)}) \tag{48}$$

which contradicts the assumption that $\tilde{\theta}_s$ is a minimizer.

Therefore, for a sufficiently small step size $\eta < \frac{2}{L}$, the updated secondary parameters $\theta_s^{(t+1)}$ are the unique minimum solution for the secondary loss function $\varphi(\theta_p^{(t)}, \theta_s)$. $\qquad\square$

**Definition A.8.** Let $f(x)$ be a function that is Lipschitz continuous with Lipschitz constant $L_f$, i.e., for any $x_1, x_2$:

$$|f(\theta_1) - f(\theta_2)| \leq L_f\|\theta_1 - \theta_2\| \tag{49}$$

**Demographic Parity Loss Function:** The demographic parity loss function $DP(f)$ is defined as:

$$DP(f) = \left|\mathbb{E}_{x \sim p(x|a=0)}[f(\theta_1; x)] - \mathbb{E}_{x \sim p(x|a=1)}[f(\theta_2; x)]\right| \tag{50}$$

where $a$ is a sensitive attribute (e.g., race, gender) with two possible values (0 and 1), and $p(x|a)$ is the conditional probability distribution of $x$ given $a$.

**Theorem A.9.** *If $f(x)$ is Lipschitz continuous with Lipschitz constant $L_f$, then the demographic parity loss function $\ell_{DP}(f)$ is also Lipschitz continuous with Lipschitz constant $L_{DP} = 2L_f$.*

*Proof.* Let $f_1(x)$ and $f_2(x)$ be two functions that are Lipschitz continuous with Lipschitz constant $L_f$. We want to show that:

$$|\ell_{DP}(f_1) - \ell_{DP}(f_2)| \leq L_{DP}\|f_1 - f_2\|_\infty \tag{51}$$

where $\|f_1 - f_2\|_\infty = \sup_x |f_1(x) - f_2(x)|$.

Consider the difference between the demographic parity loss functions:

$$|\ell_{DP}(f_1) - \ell_{DP}(f_2)| = \left|\left|\mathbb{E}_{x\sim p(x|a=0)}[f_1(x)] - \mathbb{E}_{x\sim p(x|a=1)}[f_1(x)]\right| - \tag{52}$$
$$\left|\mathbb{E}_{x\sim p(x|a=0)}[f_2(x)] - \mathbb{E}_{x\sim p(x|a=1)}[f_2(x)]\right|\right|$$
$$\leq \left|\mathbb{E}_{x\sim p(x|a=0)}[f_1(x) - f_2(x)] - \mathbb{E}_{x\sim p(x|a=1)}[f_1(x) - f_2(x)]\right| \tag{53}$$
$$\leq \mathbb{E}_{x\sim p(x|a=0)}[|f_1(x) - f_2(x)|] + \mathbb{E}_{x\sim p(x|a=1)}[|f_1(x) - f_2(x)|] \tag{54}$$
$$\leq \mathbb{E}_{x\sim p(x|a=0)}[L_f\|f_1 - f_2\|_\infty] + \mathbb{E}_{x\sim p(x|a=1)}[L_f\|f_1 - f_2\|_\infty] \tag{55}$$
$$= L_f\|f_1 - f_2\|_\infty (\mathbb{E}_{x\sim p(x|a=0)}[1] + \mathbb{E}_{x\sim p(x|a=1)}[1]) \tag{56}$$
$$= 2L_f\|f_1 - f_2\|_\infty \tag{57}$$
$$= L_{DP}\|f_1 - f_2\|_\infty \tag{58}$$

The first inequality follows from the reverse triangle inequality, the second inequality is trivial, and the third inequality follows from the Lipschitz continuity of $f_1$ and $f_2$.

Therefore, the demographic parity loss function $\ell_{DP}(f)$ is Lipschitz continuous with Lipschitz constant $L_{DP} = 2L_f$. $\qquad\square$

## A.2 Expansion to Equalized Odds (EO) difference

In Theorem A.9, we established the Lipschitz continuity of the demographic parity loss function. This approach can similarly be applied to another widely used fairness loss function, known as the equalized odds loss. The Equalized Odds Difference measures the extent to which a model's predictions deviate from equalized odds by quantifying differences in true positive rates (TPR) and false positive rates (FPR) across different groups. Mathematically, it is defined as follows:

**For true positive rate (TPR):**

$$\text{TPR Difference} = \left|\mathbb{E}_{x\sim p(x|Y=1,a=0)}[f(x)] - \mathbb{E}_{x\sim p(x|Y=1,a=1)}[f(x)]\right| \tag{59}$$

**For false positive rate (FPR):**

$$\text{TPR Difference} = \left|\mathbb{E}_{x\sim p(x|Y=0,a=0)}[f(x)] - \mathbb{E}_{x\sim p(x|Y=0,a=1)}[f(x)]\right| \tag{60}$$

The overall EO loss can then be considered as the maximum of these two differences:

$$\text{EO Difference} = \max(\text{TPR Difference}, \text{FPR Difference}) \tag{61}$$

Following the logic presented in Theorem A.9, we can determine the Lipschitz constants $L_{TPR}$ and $L_{FPR}$ for the true positive rate and false positive rate, respectively. The Lipschitz constant for the equalized odds loss can then be expressed as $\max(L_{TPR}, L_{FPR})$.

## A.3 Assumption Discussion

Our work on the FairBiNN method introduces a novel approach to addressing the fairness-accuracy trade-off in machine learning models through bilevel optimization. The theoretical foundations and empirical results demonstrate the potential of this method to outperform traditional approaches like Lagrangian regularization. However, it's crucial to examine how the underlying assumptions of our theory translate to real-world applications.

### A.3.1 Convexity Near Local Optima

One key assumption in our theoretical analysis is the convexity of the loss function near local optima. In practice, this assumption translates to the behavior of neural networks as they converge during training. While neural network loss landscapes are generally non-convex, recent research suggests that they often exhibit locally convex regions around minima, especially in overparameterized networks [3]. In real-world scenarios, as long as the network converges, it will likely encounter these locally convex regions along its optimization path. Our theory applies particularly well in these parts of the optimization process. This assumption becomes increasingly valid as the network approaches convergence, which is typically the case for well-designed models trained on suitable datasets. Therefore, practitioners can rely on this aspect of our theory as long as their models show signs of convergence on the given data.

### A.3.2 Overparameterization

The assumption of overparameterization in our model is another critical aspect that warrants discussion. In modern deep learning, overparameterized models - those with more parameters than training samples - are increasingly common. This trend aligns well with our theoretical framework. In practical terms, as long as the model is capable of overfitting on the training data, this assumption stands. This condition is often met in real-world scenarios, especially with deep neural networks applied to typical dataset sizes. The overparameterization allows for the existence of multiple solutions that can fit the training data, providing the flexibility needed for our bilevel optimization approach to find solutions that balance accuracy and fairness effectively. However, it's important to note that in some resource-constrained environments or with extremely large datasets, overparameterization might not always be feasible. In such cases, the applicability of our method may require further investigation or adaptation.

### A.3.3 Lipschitz Continuity

The assumption of Lipschitz continuity is crucial for the stability and convergence properties of our optimization process. In our experiments, we ensured that the chosen layers and loss functions satisfy Lipschitz continuity, thus upholding this assumption. For practitioners, we provide a rigorous analysis of various layers and activation functions in terms of their Lipschitz properties. This analysis serves as a guide for choosing components that maintain the Lipschitz continuity assumption. Common choices like ReLU activations and standard loss functions (e.g., cross-entropy) are Lipschitz continuous, making this assumption generally applicable in many practical scenarios. However, care must be taken when using certain architectures or custom loss functions. For instance, unbounded activation functions or poorly designed custom losses might violate this assumption. We recommend that practitioners refer to our provided analysis when designing their models to ensure compliance with this crucial property.

## A.4 Practical Implications of Bounded Output Assumption

The assumption of bounded layer outputs translates to several practical considerations in neural network design and implementation:

### A.4.1 Bounded Activation Functions

In practice, this assumption is often satisfied by using bounded activation functions:

- **Sigmoid function**: bounded between 0 and 1
- **Hyperbolic tangent (tanh)**: bounded between -1 and 1
- **ReLU6**: a variant of ReLU that is capped at 6

### A.4.2 Normalization Techniques

Various normalization techniques help ensure that the outputs of layers remain bounded:

- **Batch Normalization**: normalizes the output of a layer by adjusting and scaling the activations

- **Layer Normalization**: similar to batch normalization but normalizes across the features instead of the batch
- **Weight Normalization**: decouples the magnitude of a weight vector from its direction

### A.4.3 Regularization

Certain regularization techniques indirectly encourage bounded outputs:

- **L1/L2 regularization**: by penalizing large weights, these methods indirectly limit the magnitude of layer outputs
- **Dropout**: by randomly setting some activations to zero, dropout can help prevent excessively large outputs

By implementing these techniques, practitioners can design neural networks that better align with the theoretical assumption of bounded layer outputs. This alignment potentially leads to more stable training and improved generalization properties, bridging the gap between theoretical guarantees and practical implementations.

### A.5 Exploring Lipschitz Continuity in Activation functions

According to our theoretical framework, we can guarantee the pareto front solution when using activation functions that are Lipschitz continuous (smooth).

Activation functions ($f$) that are Lipschitz continuous have a property where there exists a constant $L$ such that for any $x, y$:

$$|f(x) - f(y)| \leq L\|x - y\| \tag{62}$$

As an example, we can prove sigmoid function is Lipschitz continuous. we need to show that there exists a constant $L$ such that for all $x, y \in \mathbb{R}$ :

$$|\sigma(x) - \sigma(y)| \leq L\|x - y\| \tag{63}$$

where $\sigma(x)$ is the sigmoid function defined as:

$$\sigma(x) = \frac{1}{1 + e^{-x}} \tag{64}$$

**Derivative of the Sigmoid Function**

The first step in proving Lipschitz continuity is to find the derivative of the sigmoid function, which gives us the rate of change. The derivative of the sigmoid function is:

$$\sigma'(x) = \sigma(x)(1 - \sigma(x)) \tag{65}$$

Since $\sigma(x)$ is always between 0 and 1, the expression $\sigma(x)(1 - \sigma(x))$ is maximized when $\sigma(x) = 0.5$.

**Finding the Lipschitz Constant**

The maximum value of $\sigma'(x)$ occurs at $\sigma(x) = 0.5$, which gives:

$$\sigma'(x) = 0.5(1 - 0.5) = 0.25 \tag{66}$$

Therefore, the derivative of the sigmoid function is bounded by 0.25:

$$0 \leq \sigma'(x) \leq 0.25 \tag{67}$$

This means that the Lipschitz constant $L$ is 0.25, and Sigmoid Function is smooth.

Similar to this proof, we can show that following common activation functions are also Lipschitz continuous:

1. **Linear:** $f(x) = x$ with constant $L = 1$

2. **Hyperbolic Tangent (Tanh):** $f(x) = \tanh(x)$ with constant $L = 1$

3. **ReLU (Rectified Linear Unit):** $f(x) = \max(0, x)$ with constant $L = 1$

4. **Leaky ReLU:** $f(x) = \max(\alpha x, x)$ where $\alpha$ is a small positive constant, with constant $L = \max(1, \alpha)$.

5. **ELU (Exponential Linear Unit):**

$$\text{ELU}(x) = \begin{cases} x, & \text{if } x > 0 \\ \alpha(e^x - 1), & \text{if } x \leq 0 \end{cases}$$

The ELU function is Lipschitz continuous, but the constant depends on the value of $\alpha$.

6. **Softplus:** $f(x) = \log(1 + e^x)$ with constant $L = 1$

There are some common activation functions that are **not** Lipschitz continuous:

1. **Softmax**

2. **Binary Step:**
$$\text{BinaryStep}(x) = \begin{cases} 1, & \text{if } x \geq 0 \\ 0, & \text{if } x < 0 \end{cases}$$

3. **Hard Tanh:**
$$\text{HardTanh}(x) = \begin{cases} -1, & \text{if } x < -1 \\ x, & \text{if } -1 \leq x \leq 1 \\ 1, & \text{if } x > 1 \end{cases}$$

4. **Hard Sigmoid:**

$$\text{HardSigmoid}(x) = \begin{cases} 0, & \text{if } x \leq -2.5 \\ 1, & \text{if } x \geq 2.5 \\ 0.2x + 0.5, & \text{if } -2.5 < x < 2.5 \end{cases}$$

### A.6 Exploring Lipschitz Continuity in CNNs and GNNs

**Convolutional Neural Networks**

The Lipschitz continuity of a CNN layer can be determined by examining its components: convolution operations, activation functions, and pooling layers. Convolution is a linear operation, and its Lipschitz constant is related to the spectral norm of the convolution matrix, typically limited by the sum of the absolute values of the weights. Activation functions like ReLU and sigmoid are Lipschitz continuous, with constants of 1 and 0.25, respectively. Pooling operations, such as max and average pooling, are also Lipschitz continuous, with max pooling having a constant of 1 and average pooling having a constant dependent on pooling size. Therefore, a CNN layer is Lipschitz continuous if all its components are, with the overall Lipschitz constant being the product of the constants of these components.

Zoe et al. [80] developed a linear programming approach to estimate the Lipschitz bound of CNN layers. Their method leverages concepts such as the Bessel bound, discrete signal processing, and the discrete Fourier transform to calculate the Lipschitz constant for each layer in popular architectures like AlexNet and GoogleNet.

**Graph Neural Networks**

Graph Neural Network (GNN) layers operate on graph-structured data through a series of message-passing, aggregation, and update steps, each contributing to the Lipschitz continuity of the layer. In the message-passing step, functions aggregate information from neighboring nodes and are often linear or involve nonlinearities; linear message-passing functions are Lipschitz continuous, with the constant depending on the weights and the graph's maximum degree. Aggregation functions, such as sum, mean, and max, are Lipschitz continuous, with sum and mean being linear, and max having a constant of 1. Update functions apply neural networks to aggregated information, and if composed

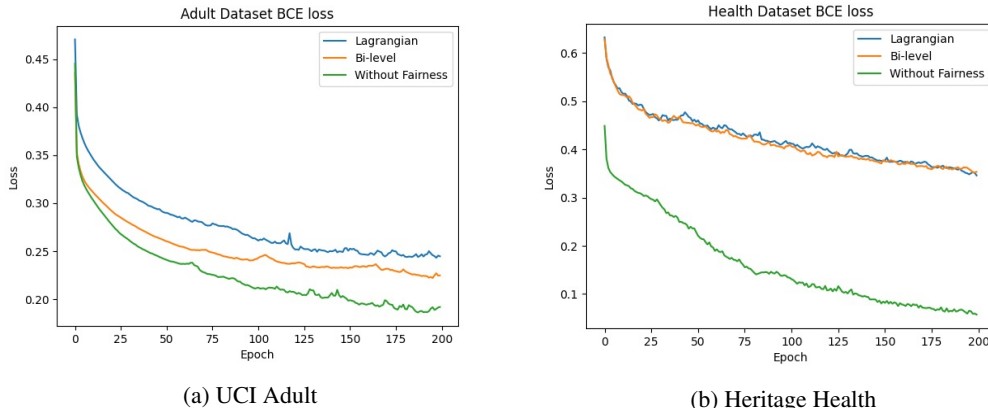

|  (a) UCI Adult | (b) Heritage Health |

Figure 2: BCE loss over epochs for the Lagrangian, Bi-level, and Without Fairness approaches on (a) the Adult dataset and (b) the Health dataset. These results illustrate that the Bi-level optimization framework achieves lower BCE loss compared to the Lagrangian approach in these experiments, highlighting its potential in optimizing both accuracy and fairness objectives in neural networks.

of Lipschitz continuous operations like linear transformations and activations such as ReLU, they maintain Lipschitz continuity. The overall Lipschitz constant of a GNN layer is influenced by the characteristics of the message-passing, aggregation, and update functions, as well as the graph's structure, such as node degrees.

A recent study by Juvina et al. [32] presents a learning framework designed to maintain tight Lipschitz-bound constraints across various GNN models. To facilitate easier computations, the authors utilize closed-form expressions of a tight Lipschitz constant and employ a constrained optimization strategy to monitor and control this constant effectively. Although this is not the first attempt to control the Lipschitz constant, the authors successfully reduce the size of the matrices involved by a factor of $K^2$, where $K$ is the number of nodes in the graph. While previous works, such as Dasoulas et al. [16], focused on controlling the Lipschitz constant for basic attention-based GNNs, Juvina et al. [32] also extend this approach to enhance the robustness of GNN models against adversarial attacks.

## A.7 Direct Comparison: Bilevel (FairBiNN) vs. Lagrangian Method

In this subsection, we present a comprehensive comparison between our proposed FairBiNN method and the traditional Lagrangian regularization approach. This comparative analysis serves multiple purposes. Primarily, it aims to empirically validate the theoretical advantages of the bilevel optimization framework outlined in our earlier analysis. By doing so, we demonstrate how the FairBiNN method translates theoretical benefits into practical performance gains in terms of both accuracy and fairness metrics. Furthermore, this comparison provides insight into the convergence behavior and stability of both methods under various hyperparameter settings. It illustrates the flexibility of the FairBiNN approach in managing the trade-off between model accuracy and fairness constraints, a crucial aspect in real-world applications of fair machine learning.

We trained both models on the Adult and Health datasets, using the same network architecture, same number of parameters, and optimization settings. Figure 2a displays the BCE loss over epochs for the Adult dataset. The Bi-level approach demonstrates better performance compared to the Lagrangian approach, achieving a lower BCE loss of approximately 0.23 by epoch 200, while the Lagrangian approach reaches a loss of about 0.26. Similarly, Figure 2b shows the BCE loss over epochs for the Lagrangian, Bi-level, and Without Fairness approaches on the Health dataset.

Through this direct comparison, we aim to bridge the gap between theoretical analysis and practical implementation, showcasing how the principled design of the FairBiNN method leads to tangible improvements in fair machine learning tasks. This offers practitioners a clear understanding of when and why they might choose the FairBiNN method over the Lagrangian approach in real-world scenarios. We trained both FairBiNN and Lagrangian models on the Adult and Health datasets, using the same network architecture (Same numeber of parameters) and optimization settings for fair

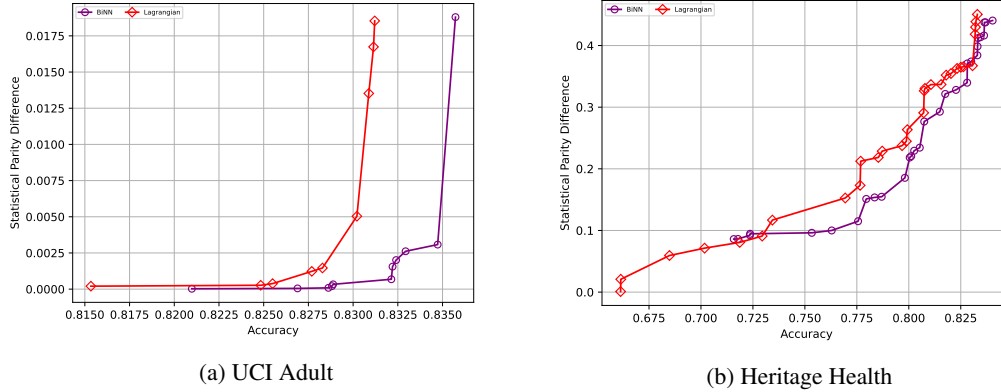

(a) UCI Adult

(b) Heritage Health

Figure 3: Comparison of FairBiNN and Lagrangian methods on UCI Adult and Heritage Health datasets

comparison. For each method, we varied the fairness-accuracy trade-off parameter ($\eta$ for FairBiNN, $\lambda$ for Lagrangian) to generate a range of models with different accuracy-fairness balances.

Figure 3 presents a comparative analysis of the FairBiNN and Lagrangian methods on two benchmark datasets: UCI Adult (Figure 3a) and Heritage Health (Figure 3b). The graphs plot the trade-off between accuracy and Statistical Parity Difference (SPD), a measure of fairness where lower values indicate better fairness. For the UCI Adult dataset (Figure 3a), we observe that the FairBiNN method consistently outperforms the Lagrangian approach. The FairBiNN curve is closer to the top-left corner, indicating that it achieves higher accuracy for any given level of fairness (SPD). The difference is particularly pronounced at lower SPD values, suggesting that FairBiNN is more effective at maintaining accuracy while enforcing stricter fairness constraints. The Heritage Health dataset results (Figure 3b) show a similar trend, but with a more dramatic difference between the two methods. The FairBiNN curve dominates the Lagrangian curve across the entire range of SPD values. This indicates that FairBiNN achieves substantially higher accuracy for any given fairness level, or equivalently, much better fairness for any given accuracy level. In both datasets, the FairBiNN method demonstrates a smoother, more consistent trade-off between accuracy and fairness. The Lagrangian method, in contrast, shows more erratic behavior, particularly in the Heritage Health dataset where its performance degrades rapidly as fairness constraints tighten. These results empirically validate the theoretical advantages of the FairBiNN method discussed earlier in the paper. They suggest that the bilevel optimization approach is more effective at balancing the competing objectives of accuracy and fairness.

### A.7.1 Computational Complexity Analysis

Let's define the following variables:

- $n$: number of parameters in $\theta_p$
- $m$: number of parameters in $\theta_s$
- $C_f$: cost of computing $f$ and its gradients
- $C_\phi$: cost of computing $\phi$ and its gradients

**Regularization (Lagrangian) Method**

The Lagrangian update rules are:

$$\theta_p = \theta_p - \alpha_L \nabla_{\theta_p}(f(\theta_p, \theta_s) + \lambda\phi(\theta_p, \theta_s)) \tag{68}$$

$$\theta_s = \theta_s - \alpha_L \nabla_{\theta_s}(f(\theta_p, \theta_s) + \lambda\phi(\theta_p, \theta_s)) \tag{69}$$

Computational complexity per iteration: $O(C_f + C_\phi + n + m)$

**Bilevel Optimization Method**

The bilevel update rules are:

$$\text{Lower level: } \theta_s = \theta_s - \alpha_s \nabla_{\theta_s} \phi(\theta_p, \theta_s) \tag{70}$$

$$\text{Upper level: } \theta_p = \theta_p - \alpha_f \nabla_{\theta_p} f(\theta_p, \theta_s^{(}\theta_p)) \tag{71}$$

Computational complexity per iteration: $O(C_f + C_\phi + n + m)$

**Empirical Comparison**

While the theoretical complexity analysis suggests similar costs for both methods, we conducted empirical tests to compare their actual runtime performance. Table 1 reports the average epoch time for both the Adult and Health datasets using the FairBiNN and Lagrangian methods after 10 epochs of warmup.

Table 1: Average epoch time (in seconds) for FairBiNN and Lagrangian methods

| Method | Adult Dataset (s) | Health Dataset (s) |
|---|---|---|
| FairBiNN | 0.62 | 1.03 |
| Lagrangian | 0.60 | 1.05 |

These experiments were conducted on an M1 Pro CPU. As we can observe from the results reported in table 1, there is no tangible difference in the average epoch time between the FairBiNN and Lagrangian methods for both datasets. This empirical evidence aligns with our theoretical analysis.

## A.8 Related works - Graph and Vision domains

### A.8.1 Graph

The message-passing structure of GNNs and the topology of graphs both have the potential to amplify the bias. In general, in graphs such as social networks, nodes with sensitive features similar to one another are more likely to link to one another than nodes with sensitive attributes dissimilar from one another [18, 52]. On social networks, for instance, persons of younger generations have a higher tendency to form friendships with others of a similar age [18]. This results in the aggregation of neighbors' features in GNN having similar representations for nodes of similar sensitive information while having different representations for nodes of different sensitive features, which leads to severe bias in decision-making, in the sense that the predictions are highly correlated with the sensitive attributes of the nodes. GNNs have a greater bias due to the adoption of graph structure than models that employ node characteristics [14]. Because of this bias, the widespread use of GNNs in areas such as the evaluation of job candidates [44] and the prediction of drug-target interaction [72, 36] would be significantly hindered. As a result, it is essential to research equitable GNNs. The absence of sensitive information presents significant problems to the work that has already been done on fair models [5, 13, 39, 41, 64]. Despite the significant amount of work that has been put into developing fair models through the revision of features [33, 34, 78], disentanglement [13, 41], adversarial debiasing [5, 21], and fairness constraints [74, 75], these models are almost exclusively designed for independently and identically distributed (i.i.d) data, meaning that they are unable to be directly applied to graph data due to the fact that they do not simultaneously take into consideration the bias that comes from node attributes and graph. In recent years, Bose and Hamilton [6], Rahman et al. [52] have been published to learn fair node representations from graphs. These approaches only deal with simple networks that do not have any properties on any of the nodes, and they place their emphasis on fair node representations rather than fair node classifications. Finally, Dai and Wang [14] used graph topologies and a restricted amount of protected attributes and designed FairGNN to reduce the bias of GNNs while retaining high node classification accuracy.

### A.8.2 Vision

The challenges caused by bias in computer vision might appear in various ways. It has been found, for instance, that in action recognition models, when the data include gender bias, the bias is exacerbated by the models trained on such datasets [79]. Face detection and recognition models may be less precise for some racial and gender categories [7]. Methods for mitigating bias in vision datasets are suggested in [65] and [71]. Several researchers have used GANs on image datasets for bias reduction. Sattigeri et al. [57] altered the utility function of GAN in order to generate equitable picture

Table 2: Summary of Parameter Setting for the fairness layers on tabular datasets

| Hyperparameters | UCI Adult | Health Heritage |
|---|---|---|
| FC layers before the fairness layers | 2 | 2 |
| Fairness FC layers | 1 | 3 |
| FC layers after the fairness layers | 1 | 1 |
| Epoch | 50 | 50 |
| Batch size | 100 | 100 |
| Dropout | 0 | 0 |
| Network optimizer | Adam | Adam |
| fairness layers' optimizer | Adam | Adam |
| classifier layers' learning rate | 1e-3 | 1e-3 |
| fairness layers' learning rate | 1e-5 | 1e-5 |
| $\eta$ | 100 | 100 |

datasets. FairFaceGAN [29] provides facial image-to-image translation to avoid unintended transfer of protected characteristics. Roy and Boddeti [56] developed a method to mitigate information leakage on image datasets by formulating the problem as an adversarial game to maximize data utility and minimize the amount of information contained in the embedding, measured by entropy. Ramaswamy et al. [55] presents a methodology to generate balanced training data for each protected property by perturbing the latent vector of a GAN. Other experiments using GANs to generate accurate data are [8, 58]. Beyond GANs, many strategies have addressed the challenge of AI fairness. [54] proposed a U-Net for creating unbiased image data. Deep information maximization adaption networks were employed to eliminate racial bias in face vision datasets [67], while reinforcement learning was utilized for training a race-balanced network [66]. Wang et al. [69] offer a generative few-shot cross-domain adaptation method for performing fair cross-domain adaptation and enhancing minority category performance. The research in [70] recommends adding a penalty term to the softmax loss function to reduce bias and enhance face recognition fairness performance. Quadrianto et al. [51] describes a technique for discovering fair data representations with the same semantic information as the original data. There have also been effective applications of adversarial learning for this purpose [68, 77]. [9] proposed fair mixup, which uses data augmentation to mitigate bias in data.

### A.9 Evaluation Metrics

We utilize four metrics to compare our model's performance against baseline models. Average precision (AP) is utilized to gauge classifier accuracy, combining recall and accuracy at each point. In tabular and graph datasets, we opt for accuracy to align with existing literature practices.

Fairness evaluation draws from various criteria, with demographic parity (DP) being widely used. DP quantifies the difference in favorable outcomes across protected groups, expressed as ($|P(Y = 1|S = 0) - P(Y = 1|S = 1)|$) [44]. For scenarios involving more than two groups, DP can be calculated as $\Delta_{DP}(a, \hat{y}) = \max_{a_i, a_j} |P(\hat{y} = 1|a = a_i) - P(\hat{y} = 1|a = a_j))|$ [25]. A smaller DP indicates fairer categorization. We also adopt the difference in equality of opportunity ($\Delta$EO), which measures the absolute difference in true positive rates between gender expressions ($|TPR(S = 0) - TPR(S = 1)|$) [40, 55]. Minimizing $\Delta$EO signifies fairer outcomes. Demographic parity serves as the fairness criterion in our optimization. Discrepancies between EO and DP may occur due to this choice.

### A.10 Implementation details

The hyperparameters used in training the models on each dataset can be found in the tables 2, 3, and 4. The training was conducted on a computer with an NVIDIA GeForce RTX 3090.

Table 3: Summary of Parameter Setting for the fairness layers on graph datasets

| Hyperparameters | POKEC-Z | POKEC-N | NBA |
|---|---|---|---|
| GCN layer before the fairness layers | 2 | 2 | 2 |
| Fairness FC layers | 1 | 1 | 1 |
| FC layers after the fairness layers | 1 | 1 | 1 |
| Epoch | 5000 | 1000 | 1000 |
| Batch size | 1 | 1 | 1 |
| Dropout | 0 | 0.5 | 0.5 |
| Network optimizer | Adam | Adam | Adam |
| fairness layers' optimizer | Adam | Adam | Adam |
| classifier layers' learning rate | 1e-3 | 1e-3 | 1e-2 |
| fairness layers' learning rate | 1e-6 | 1e-8 | 1e-5 |
| $\eta$ | 1000 | 100 | 1000 |

Table 4: Summary of Parameter Setting for the fairness layers on vision dataset

| Hyperparameters | CelebA-Attractive | CelebA-Smiling | CelebA-WavyHair |
|---|---|---|---|
| Fairness FC layers | 1 | 1 | 1 |
| FC layers after the fairness layers | 1 | 1 | 1 |
| Epoch | 30 | 15 | 15 |
| Batch size | 128 | 128 | 128 |
| Dropout | 0 | 0 | 0 |
| Network optimizer | Adam | Adam | Adam |
| fairness layers' optimizer | Adam | Adam | Adam |
| classifier layers' learning rate | 1e-3 | 1e-3 | 1e-3 |
| fairness layers' learning rate | 1e-6 | 1e-5 | 1e-5 |
| $\eta$ | 1000 | 100 | 100 |

## A.11 Other domains' results

### A.11.1 Graph

We compare our suggested framework with some of the cutting-edge approaches for fair classification, and fair graph embedding learning including ALFR [21], ALFR-e, Debias [77], Debias-e, and FCGE [6]. In the ALFR [21] approach, which is a pre-processing technique, the sensitive information in the representations created by an MLP-based autoencoder is eliminated using a discriminator. Then the debiased representations are used to train the linear classifier. ALFR-e is a method to make use of the graph structure information and joins the user features in the ALFR with the graph embeddings discovered by deepwalk [48]. Debias [77], is a fair categorization technique used throughout processing. It immediately applies a discriminator to the predicted likelihood of the classifier. Debias-e, which is similar to the ALFR-e, also includes deepwalk embeddings into the Debias characteristics. FCGE [6], is suggested as a method for learning fair node embeddings in graphs without node characteristics. FairGCN [14], a graph convolutional network designed for fairness in graph-based learning. It incorporates fairness constraints during training to reduce disparities between protected groups. FairGAT [37], a fairness-aware graph-based learning framework that employs a novel attention learning strategy to reduce bias. This framework is grounded in a theoretical analysis that identifies sources of bias in GAT-based neural networks used for node classification. NT-FAIRGNN [15] is a graph neural network that aims to achieve fairness by balancing the trade-off between accuracy and fairness. It uses a two-player minimax game between the predictor and the adversary, where the adversary aims to maximize the unfairness. Discriminators screen out the delicate data in the embeddings. We used the Dai and Wang [14] study's obtained datasets for our investigation which are as follows:

*Pokec* [62] is among the most well-known social network datasets in Slovakia which resemble Facebook and Twitter greatly. This dataset includes anonymous information from the whole social network of the year 2012. User profiles on Pokec include information on gender, age, interests, hobbies, profession, and more. There are millions of users in the original Pokec dataset. Sampled

Table 5: The comparisons of our proposed method with the baselines on Pokec-z

| METHOD | ACC(%) | AUC(%) | $\Delta_{DP}$(%) | $\Delta_{EO}$(%) |
|---|---|---|---|---|
| ALFR [21] | 65.4 ±0.3 | 71.3 ±0.3 | 2.8 ±0.5 | 1.1 ±0.4 |
| ALFR-E [21, 48] | 68.0 ±0.6 | 74.0 ±0.7 | 5.8 ±0.4 | 2.8 ±0.8 |
| DEBIAS [77] | 65.2 ±0.7 | 71.4 ±0.6 | 1.9 ±0.6 | 1.9 ±0.4 |
| DEBIAS-E [77, 48] | 67.5 ±0.7 | 74.2 ±0.7 | 4.7 ±1.0 | 3.0 ±1.4 |
| FCGE [6] | 65.9 ±0.2 | 71.0 ±0.2 | 3.1 ±0.5 | 1.7 ±0.6 |
| FAIRGCN [14] | 70.0 ±0.3 | 76.7 ±0.2 | 0.9 ±0.5 | 1.7 ±0.2 |
| FAIRGAT [37] | 70.1 ±0.1 | 76.5 ±0.2 | **0.5 ±0.3** | **0.8 ±0.3** |
| NT-FAIRGNN [15] | 70.0 ±0.1 | 76.7 ±0.3 | 1.0 ±0.4 | 1.6 ±0.2 |
| **GAT+FAIRBINN (OURS)** | **70.97 ±0.16** | **77.58 ±0.13** | 0.93 ±0.44 | 0.97 ±0.40 |

Table 6: The comparisons of our proposed method with the baselines on Pokec-n

| METHOD | ACC(%) | AUC(%) | $\Delta_{DP}$(%) | $\Delta_{EO}$(%) |
|---|---|---|---|---|
| ALFR [21] | 63.1 ±0.6 | 67.7 ±0.5 | 3.05 ±0.5 | 3.9 ±0.6 |
| ALFR-E [21, 48] | 66.2 ±0.5 | 71.9 ±0.3 | 4.1 ±0.5 | 4.6 ±1.6 |
| DEBIAS [77] | 62.6 ±0.9 | 67.9 ±0.7 | 2.4 ±0.7 | 2.6 ±1.0 |
| DEBIAS-E [77, 48] | 65.6 ±0.8 | 71.7 ±0.7 | 3.6 ±0.2 | 4.4 ±1.2 |
| FCGE [6] | 64.8 ±0.5 | 69.5 ±0.4 | 4.1 ±0.8 | 5.5 ±0.9 |
| FAIRGCN [14] | 70.1 ±0.2 | 74.9 ±0.4 | 0.8 ±0.2 | 1.1 ±0.5 |
| FAIRGAT [37] | 70.0 ±0.2 | 74.9 ±0.4 | **0.6 ±0.3** | **0.8 ±0.2** |
| NT-FAIRGNN [15] | **70.1 ±0.2** | 74.9 ±0.4 | 0.8 ±0.2 | 1.1 ±0.3 |
| **GAT+FAIRBINN (OURS)** | 70.07 ±0.5 | **75.8 ±0.38** | 0.62 ±0.14 | 3.0 ±1.0 |

Pokec-z and Pokec-n datasets are based on the provinces that users are from. The categorization task involves predicting the users' working environment.

*NBA* is a Kaggle dataset with about 400 NBA basketball players that served as the basis for this extension. Players' 2016–2017 season success statistics, along with additional details like nationality, age, and income are presented. They gathered the relationships between NBA basketball players on Twitter using its official crawling API to create the graph connecting the NBA players. They separated the nationality into two groups, American players and international players, which is a sensitive characteristic. The classification job is to predict whether a player's wage is above the median.

Each experiment was conducted five times, and Tables 5, 6, and 7 report the mean and standard deviation of the runs for Pokec-z, Pokec-n, and NBA datasets, respectively. These results represent the selected Pareto solutions for comparison with the benchmarks. The tables reveal that, in comparison to GAT, generic fair classification techniques and graph embedding learning approaches exhibit inferior classification performance, even when utilizing graph information. In contrast, our Bilevel design performs comparably to baseline GNNs. FairGCN is close to the baseline, but the FairBiNN approach outperforms it. When sensitive information is scarce (e.g., NBA dataset), baselines exhibit clear bias, with graph-based baselines performing worse. However, our proposed model yields near-zero statistical demographic parity, indicating effective discrimination mitigation.

### A.11.2  Vision

We compare our method on vision task with (1) empirical risk minimization (ERM), which accomplishes training task without any regularization, (2) gap regularization, which directly regularizes the model, (3) adversarial debiasing [77], and (4) Fairmixup [9]. To showcase the effectiveness of our method, we employed the CelebA dataset of face attributes [38], which comprises over 200,000 images of celebrities. Each image in the dataset has been assigned 40 binary attributes, including gender, that were labeled by humans. We selected the attributes of attractive, smiling, and wavy hair and used them in three binary classification tasks, with gender serving as the protected attribute. The reason we chose these three attributes is that each of them has a group that is sensitive to them and receives a disproportionately high number of positive samples. We trained a ResNet-18 model [27]

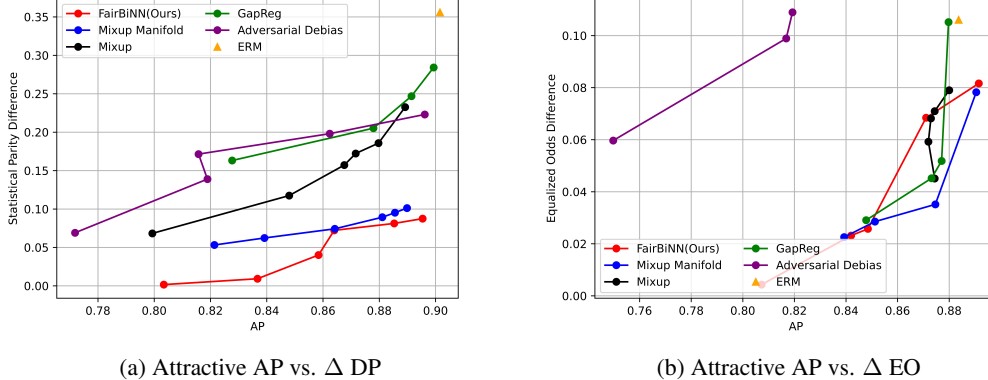

(a) Attractive AP vs. $\Delta$ DP

(b) Attractive AP vs. $\Delta$ EO

Figure 4: *Attractive Attribute of CelebA Dataset as the Target Attribute*. (a) reflects the trade-off between Average Precision and Demographic Parity Difference. (b) shows the trade-off between Average Precision and Equalized Odds Difference.

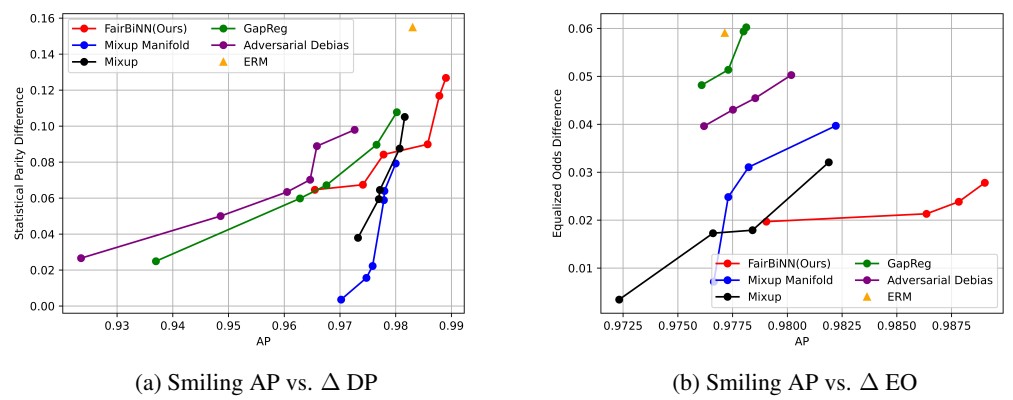

(a) Smiling AP vs. $\Delta$ DP

(b) Smiling AP vs. $\Delta$ EO

Figure 5: *Smiling Attribute of CelebA Dataset as the Target Attribute*. (a) reflects the trade-off between Average Precision and Demographic Parity Difference. (b) shows the trade-off between Average Precision and Equalized Odds Difference.

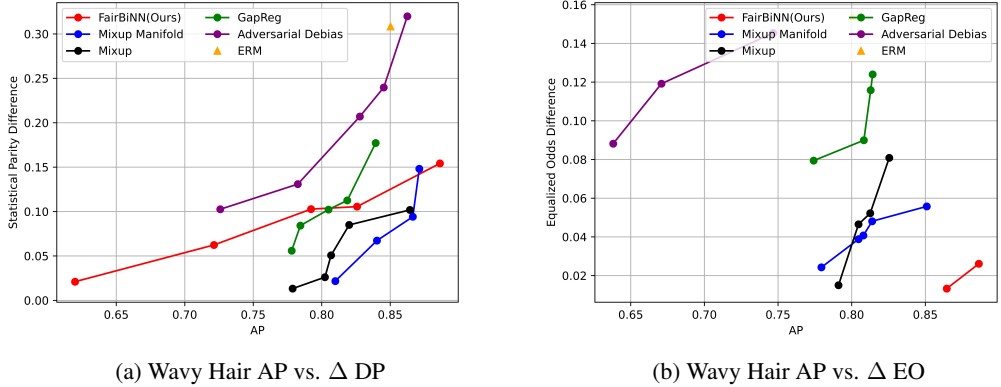

(a) Wavy Hair AP vs. $\Delta$ DP

(b) Wavy Hair AP vs. $\Delta$ EO

Figure 6: *Wavy Hair Attribute of CelebA Dataset as the Target Attribute*. (a) reflects the trade-off between Average Precision and Demographic Parity Difference. (b) shows the trade-off between Average Precision and Equalized Odds Difference. The FairBiNN method is showing competitive results to the baseline.

Table 7: The comparisons of our proposed method with the baselines on NBA

| METHOD | ACC(%) | AUC(%) | $\Delta_{DP}$(%) | $\Delta_{EO}$(%) |
|---|---|---|---|---|
| ALFR [21] | 64.3 ±1.3 | 71.5 ±0.3 | 2.3 ±0.9 | 3.2 ±1.5 |
| ALFR-E [21, 48] | 66.0 ±0.4 | 72.9 ±1.0 | 4.7 ±1.8 | 4.7 ±1.7 |
| DEBIAS [77] | 63.1 ±1.1 | 71.3 ±0.7 | 2.5 ±1.5 | 3.1 ±1.9 |
| DEBIAS-E [77, 48] | 65.6 ±2.4 | 72.9 ±1.2 | 5.3 ±0.9 | 3.1 ±1.3 |
| FCGE [6] | 66.0 ±1.5 | 73.6 ±1.5 | 2.9 ±1.0 | 3.0 ±1.2 |
| FAIRGCN [14] | 71.1 ±1.0 | 77.0 ±0.3 | 1.0 ±0.5 | 1.2 ±0.4 |
| FAIRGAT [37] | 71.5 ±0.8 | 77.5 ±0.7 | 0.7 ±0.5 | **0.7 ±0.3** |
| NT-FAIRGNN [15] | 71.1 ±1.0 | 77.0 ±0.3 | 1.0 ±0.5 | 1.2 ±0.4 |
| **GAT+FAIRBiNN (OURS)** | **77.09 ±0.45** | **77.99 ±0.58** | **0.34 ±0.21** | 12.78 ±2.9 |

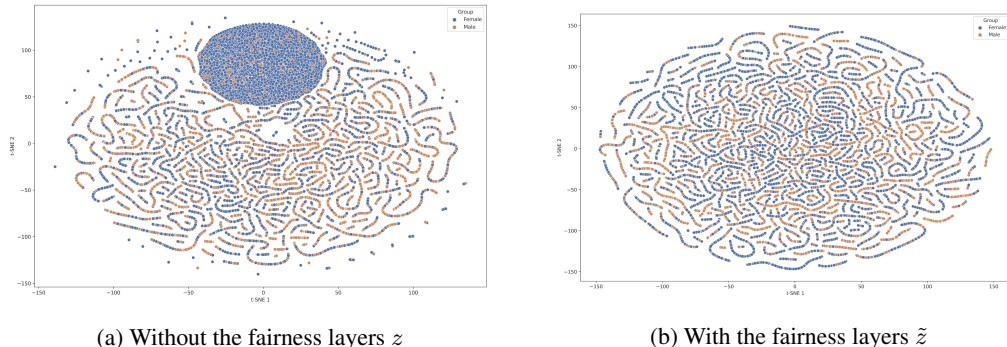

(a) Without the fairness layers $z$          (b) With the fairness layers $\tilde{z}$

Figure 7: CelebA Dataset – t-SNE visualization of $z$ and $\tilde{z}$ labeled with gender classes. The invariant encoding $\tilde{z}$ shows no clustering by gender. These plots are generated using attractive attribute.

for each task and added two additional layers to predict the outcomes. The trade-off between Average Precision (AP), Demographic Parity (DP), and Equality of Opportunity (EO) for attributes "Attractive", "Smiling", and "Wavy Hair" is illustrated in the figures 4, 5, and 6 respectively. Our proposed method provides a more balanced trade-off between accuracy and fairness. Instead of prioritizing one over the other, our method strikes a better balance, ensuring that the trained model is both accurate and fair. Moreover, the FairBiNN model consistently provides better equality of opportunity across various accuracy levels compared to benchmark models. Through empirical validation on multiple benchmarks, we've shown that the FairBiNN approach consistently outperforms other methods in achieving equality of opportunity across various accuracy levels. This indicates that our method can provide fair treatment to different protected groups while still maintaining high predictive accuracy.

Furthermore, we demonstrate the power of the bilevel design for fairness by visualizing the t-SNE plot. The t-SNE visualization of $z$ (output of the ResNet-18 before the classification layer without the fairness layers) and $\tilde{z}$ (output of the ResNet-18 before the classification layer with the Bilevel fairness) are shown in Figures 7a and 7b, demonstrating that $z$ clusters by gender, but $\tilde{z}$ does not. Further insights about ablation study outcomes are detailed in section A.13.

## A.12 Architecture visualization

To better illustrate the training process outlined in Algorithm 1, we present the network architecture in Figure 8.

## A.13 Ablation Study

### A.13.1 Impact of $\eta$ on Model Performance

To understand the sensitivity of our FairBiNN model to the choice of $\eta$, we conducted an ablation study on both the Adult and Health datasets. The parameter $\eta$ controls the trade-off between accuracy and fairness in our bilevel optimization framework.

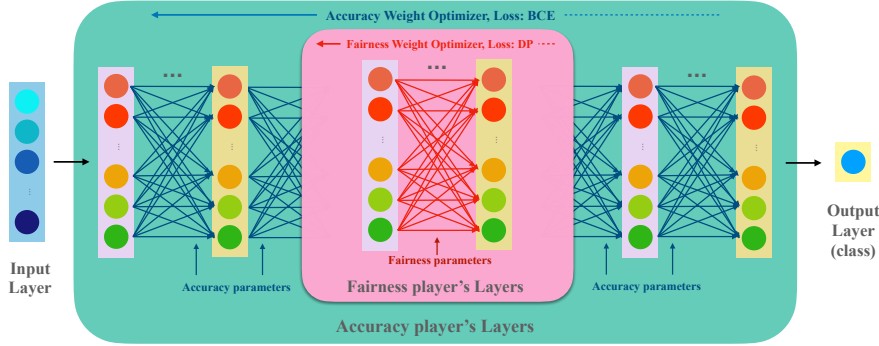

Figure 8: The FairBiNN network architecture illustrating the process described in Algorithm 1.

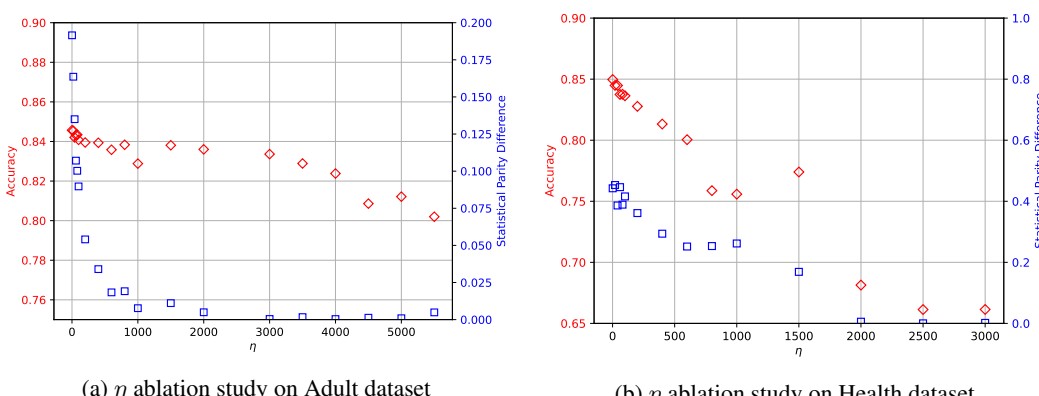

(a) $\eta$ ablation study on Adult dataset

(b) $\eta$ ablation study on Health dataset

Figure 9: Ablation study on the impact of $\eta$ parameter across two datasets. (a) Results on the Adult dataset showing the effect of different $\eta$ values. (b) Similar analysis conducted on the Health dataset, demonstrating how $\eta$ influences model performance.

### A.13.2 Experimental Setup

We varied $\eta$ across a range of values: 1-6000 for Adult dataset, and 1-3000 for Health dataset. For each value of $\eta$, we trained the FairBiNN model on both the Adult and Health datasets, keeping all other hyperparameters constant. We evaluated the models based on accuracy and demographic parity (DP).

### A.13.3 Results

Figures 9a and 9b show the results of our ablation study for the Adult and Health datasets, respectively. The results demonstrate a clear trade-off between accuracy and fairness as $\eta$ varies. For both datasets: As $\eta$ increases, the demographic parity (DP) decreases, indicating improved fairness. However, this improvement in fairness comes at the cost of reduced accuracy. The relationship is not linear; there are diminishing returns in fairness improvement as $\eta$ increases, especially at higher values. For the Adult dataset, setting $\eta = 1000$ appears to offer a good balance, achieving a DP of 0.012 while maintaining an accuracy of 82.9%. For the Health dataset, $\eta = 700$ also provides a reasonable trade-off with a DP of 0.23 and an accuracy of 80.2%. These results highlight the importance of carefully tuning $\eta$ to achieve the desired balance between accuracy and fairness. The optimal value may vary depending on the specific requirements of the application and the characteristics of the dataset.

This ablation study demonstrates that our FairBiNN model provides a flexible framework for managing the accuracy-fairness trade-off through the $\eta$ parameter. Practitioners can adjust $\eta$ based on their specific fairness requirements and acceptable accuracy thresholds. Future work could explore adaptive schemes for setting $\eta$ during training to automatically find an optimal balance.

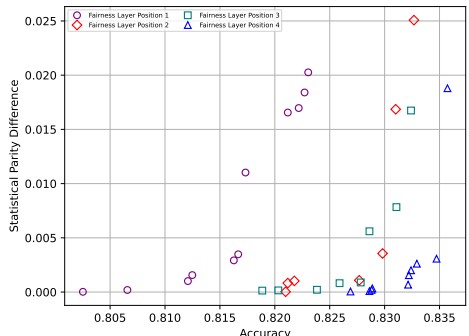 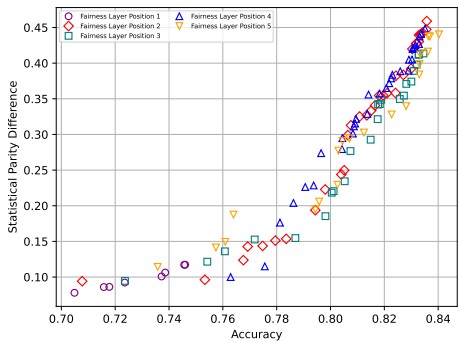

(a) Fairness layer position ablation study on Adult dataset

(b) Fairness layer position ablation study on Health dataset

Figure 10: Fairness Layers Position ($i$), where $i$ indicates the $i - th$ hidden layer

**Position of Fairness Layers** To understand the impact of the position of fairness layers within the network architecture, we conducted an ablation study varying their placement. This study aims to identify the optimal position for fairness layers and provide insights into why certain positions may be more effective.

**Experimental Setup**

We tested 4 fairness layer positions on the Adult dataset and 5 fairness layer positions on the Health dataset. For each configuration, we kept the total number of parameters constant to ensure a fair comparison. We evaluated the models based on accuracy and demographic parity (DP).

**Results** Figures 10a and 10b show the results of our ablation study for the Adult and Health datasets, respectively.
The results consistently show that placing the fairness layers just before the output layer (in the last hidden layer) yields the best performance in terms of both accuracy and fairness. This configuration achieves the highest accuracy while maintaining the lowest demographic parity on both datasets.

Several factors contribute to the superior performance of fairness layers when placed in the last hidden layer:

- **Rich Feature Representations**: By the time the data reaches the last hidden layer, the network has already learned rich, high-level feature representations. This allows the fairness layers to operate on more informative features, potentially making it easier to identify and mitigate biases.
- **Minimal Information Loss**: Placing fairness layers earlier in the network might lead to loss of important information that could be useful for the classification task. By positioning them at the end, we ensure that all relevant features are preserved throughout most of the network.
- **Direct Influence on Output**: Being closest to the output layer, fairness layers in this position have the most direct influence on the final predictions. This allows for more effective bias mitigation without excessively disturbing the learned representations in earlier layers.
- **Gradient Flow**: In backpropagation, gradients from the fairness objective have a shorter path to travel when the fairness layers are near the output. This might lead to more stable and effective updates for bias mitigation.
- **Adaptability**: Fairness layers at the end of the network can adapt to various biases that might emerge from complex interactions in earlier layers, providing a final "correction" before the output.

### A.13.4 Number and Type of Fairness Layers

In this subsection, we perform an ablation study to investigate the effects of different functions for the fairness layers. The fairness layer can be any differentiable function with controllable parameters

Table 8: Area over the curve of statistical demographic parity and accuracy for model ablation

| METHOD | UCI ADULT | HERITAGE HEALTH |
|---|---|---|
| ONE LINEAR LAYER | **0.411** | 0.492 |
| TWO LINEAR LAYERS | 0.404 | 0.513 |
| THREE LINEAR LAYERS | 0.349 | **0.531** |

denoted as $\theta_d$. We experimented with three configurations for the fairness layers: one linear layer, two linear layers, and three linear layers on tabular datasets. The results of the ablation study are summarized in Table 8.

For the CelebA dataset, we explored three types of fairness layers: linear layers, Residual Blocks (ResBlocks), and Convolutional Neural Network (CNN) layers. The mean scores of each category of CelebA attributes for each type of fairness layer are provided in Table 9.

The justification for the performance differences between the ResBlock and the fully connected models in our ablation study lies in the proportion of the model occupied by the fairness layers and the specific contributions of these layers to different parts of the network. In particular, there are two primary factors that explain the observed performance differences:

- **Role in the Network**: The ResBlock and the fully connected modules serve different purposes within the network. The ResBlock contributes to the embedding space of the image, which includes feature extraction and representation learning. This enables the model to capture the essential characteristics of the image while minimizing the effect of the protected attributes (e.g., gender) on the classification task. In contrast, the fully connected module is mainly involved in the classification part of the network, where it contributes to the decision-making process based on the features extracted from the previous layers. This distinction in roles explains why the ResBlock provides more fair results, as it directly affects the representation learning and reduces the influence of the protected attributes on the embeddings.

- **Flow of Data**: The flow of data through the ResBlock is different from the flow through the fully connected and CNN modules. ResBlocks have skip connections that allow the input to bypass some layers and directly flow to the subsequent layers. These skip connections help in preserving the original information and preventing the loss of critical features during the network's forward pass. As a result, the ResBlock is more effective in capturing the inherent relationships in the data while mitigating the bias from the protected attributes [28]. In contrast, CNNs involve multiple convolution and pooling operations, which can cause the loss of some information relevant to fairness. The fully connected module, with its dense layers, lacks the skip connections present in the ResBlock, which can lead to less effective bias mitigation.

In conclusion, our ablation study demonstrates that the choice of layer in the fairness layers can significantly impact the fairness and accuracy of the model. It is essential to strike a balance between fairness and accuracy and to select the appropriate fairness layer for the specific dataset and application at hand.

Table 9: Accumulative comparison between different fairness layers

| CNNBlock | AP | $\Delta$DP | $\Delta$EO |
|---|---|---|---|
| One Linear Layer | **0.646** | 0.072 | **0.084** |
| CNN Res Block | 0.568 | **0.04** | 0.126 |
| CNN Layer | 0.617 | 0.058 | 0.099 |

## A.14 Ethical & Broader Social Impact

This work introduces a novel bilevel optimization framework for multi-objective optimization in neural networks. While we use fairness as a case study, it's important to note that our method is not inherently a fairness research technique, but rather a general optimization approach that can be applied to various secondary objectives.

In the context of our fairness case study, our FairBiNN method shows promising results in optimizing the trade-off between demographic parity (a measure of group fairness) and accuracy. While these results are encouraging, it is crucial to consider the broader ethical implications and potential societal impacts of applying this technique to fairness problems.

On the positive side, when applied to fairness, our approach could help reduce discriminatory outcomes in high-stakes automated decision making systems, promoting more equitable treatment across protected groups in domains like hiring, lending, and healthcare [44, 76]. By providing a flexible framework to manage the accuracy-fairness trade-off, practitioners can fine-tune models to meet specific fairness requirements mandated by regulations or organizational policies.

However, we must also consider potential negative consequences. There is a risk that mathematical notions of fairness like demographic parity could provide a false sense of ethical assurance, when fairness is a complex social and philosophical concept that cannot be fully captured by simple statistical measures [20, 50]. Our focus on group fairness, while important, does not guarantee individual fairness [73].

The mathematical formulation we present, while rigorous, should not be seen as providing absolute ethical guarantees when applied to fairness problems. Over-reliance on our method without careful consideration of the broader context could lead to unintended harms [9, 39].

As with any machine learning technique, it is the responsibility of practitioners to properly configure and test models for their specific use cases. Our method provides additional tools for optimization but does not absolve practitioners of their ethical obligations. We believe that providing more precise control over trade-offs actually enables more ethical implementations. However, practitioners must be aware that while our approach performs well in our experiments, particularly in balancing demographic parity and accuracy, real-world applications may present unforeseen challenges and edge cases [31, 49].

Ultimately, while our work provides a useful tool for multi-objective optimization in neural networks, its application to fairness should not be seen as a complete solution to algorithmic bias. Continued interdisciplinary collaboration between computer scientists, ethicists, policymakers and impacted communities is essential to develop AI systems that are truly fair and beneficial to society [6, 60, 71].

