# OpenReview forum: "Fair Bilevel Neural Network (FairBiNN): On Balancing fairness and accuracy via Stackelberg Equilibrium"
_NeurIPS.cc/2024/Conference — NeurIPS 2024 poster_

### Official Review · Reviewer_dbnD · 2024-06-17

**Soundness:** 3
**Presentation:** 2
**Contribution:** 3
**Rating:** 5
**Confidence:** 4

**Summary:**

The authors propose a bi-level optimization approach to simultaneously optimize a performance and a fairness loss. They show that this approach is, in theory, superior to a typical approach that regularizes the performance objective with a fairness objective. Using tabular, graph and vision datasets, they demonstrate the Pareto frontiers obtained by their method, compared to multiple (mostly regularization) baselines.

**Strengths:**

Regularization and adversarial training can be difficult to perform in practice, as the choice of hyper-parameters needs to be performed with care and convergence might not be guaranteed. By leveraging bi-level optimization, the authors provide a novel perspective on the problem (to the best of my knowledge).

**Weaknesses:**

While I found the idea interesting, I found that the paper lacked clarity, and have a few major comments / concerns at this stage.

**Major concerns**

- I found that there were many assumptions made, basically requiring that all parameters are smooth, change at a limited rate and improve compared to prior steps. While some of these might be reasonable, the paper lacked a discussion of what these assumptions mean, and in which conditions they might not be respected.
- I am not convinced by Theorem 9 and its proof, as it relies on two functions but DP is computed on a single function (see below).
- I found that the connection between the theory and its implementation is not well detailed and explored.
- To me, discussing works that perform constrained or multi-objective optimization of hyper-parameters, or refer to meta-learning for fairness would be relevant.
- Practically, I see multiple obstacles to the adoption of the proposed method: more hyper-parameters to select, published models cannot be re-used, only one fairness constraint is considered, ... These are not discussed or acknowledged.

**Detailed comments**:
- Related works: I believe the works who perform the multi-objective or constrained optimization of model’s hyperparameters (e.g. Perrone et al., 2020, https://arxiv.org/pdf/2006.05109 and many others) should be discussed, and potentially considered as baselines. While I understand they do not provide the same guarantees, they are easier to implement.
- Can the authors discuss their approach with e.g. Slack et al., 2019 (https://arxiv.org/abs/1911.04336), in which the 2 objectives are separated into learners and a meta-learner instead?
- The related works section describes some related works with one sentence, but does not “group” them into the different avenues they represent, or how they differ from the proposed approach. I would suggest trying to “analyze” the field at a higher level and show what the proposed approach can bring.
- Overall I found the methodology section dense and not easy to follow. The assumptions are also not discussed, simply enumerated and it is unclear how realistic they are.
- The experimental baselines seem to encompass multiple techniques, but are not well described. It is unclear how each of these compares to the proposed approach.
- For the Health dataset, as well as for other datasets (in Appendix), the theoretical results do not seem to hold. This is briefly discussed at the end of the results section, but is a clear limitation of the work. There is also little description of whether the selection of hyper-parameters is more complex for the bi-level case.
- Relatedly, this is presented as an advantage but to me is a weakness of the approach that the architecture needs to be completely revised to include fairness layers. This means that published models cannot easily be reused, and that the whole process needs to be re-trained for a different fairness constraint.

**Questions:**

- Line 550: how can ensure that $\hat{\theta}_p$ is closer to the optimum than $\theta_p$? Strict improvement at each step seems like a strong assumption to me. Shouldn’t this be spelled out as an extra assumption?
- How does the assumption of “sufficiently small” $\eta$ maps to the practical implementation, as we do not know the Lipschitz constant?
- Similarly, how about the assumption in Theorem 9 of an overparameterization?
- Line 212, DP: why use $\theta_1$ and $\theta_2$? From Assumption 3.10, it suggests that these represent the parameters for $x_1$ and $x_2$. However, the parameters $\theta$ when computing DP are the same for $a=0$ and $a=1$. I don’t see how we can guarantee that the network’s activations are Lipschitz continuous unless we also make assumptions of the distance between input examples (which can be difficult to estimate). The proof also refers to 2 functions $f_1$ and $f_2$, but we have the same function, only different subsets of inputs $x_{a=0}$ and $x_{a=1}$. Can you clarify please? It is not clear to me how the proof shows that DP is Lipschitz continuous.
- The discussion seems more like a brief summary with a rebuttal paragraph, rather than a proper discussion. The work is not put in perspective with the literature.
- Figures 4 and 5 in Appendix: the proposed method is not strictly better compared to the baselines in all cases. Is there a pattern of failure cases?

**Minor**:

- Line 64: is the comment pointing towards distribution shifts in the test data? It is unclear what “depends on the data” means.
- “Paractical” line 198
- $\hat{\theta}$ is not defined in Assumption 3.3.
- Equation (1): the description uses the ‘p’ and ‘s’ subscripts while the equation uses ‘a’ and ‘f’. Please correct.
- Is Assumption 3.10 a repeat of definition 3.1?
- Line 276: end with “shows”, I am assuming something is missing.
- Line 545, equation 38: shouldn’t it be $\phi$ instead of f?
- Table 4 (Appendix): it looks like FairGAT has smaller delta EO and should be highlighted instead. Overall, I’d suggest highlighting all results within the standard deviation of each other, as it is a bit misleading otherwise.

**Limitations:**

The authors describe as “NA” the limitations of their approach. I strongly believe that all methods have limitations and these should be discussed.
- For instance, multiple assumptions are made, but not discussed.
- It is also unclear how the practical implementation reflects the theoretical results, as it would be reasonable to expect effects of finite sample size, the hyper-parameters selected for each of the fairness / performance layers, …
- Are any common activation functions not Lipschitz continuous? What about ELU, or other fairness criteria? What would that mean for the derived results?

---

> ### Author Rebuttal · Authors · 2024-08-06
>
> Thank you for your thoughtful feedback. We have addressed your questions and concerns below. If you have any further questions, we would be glad to discuss them.
> ## Weaknesses
> ### W1
> We appreciate the reviewer's feedback on the assumptions in our paper. In response, we have added a dedicated discussion section in global rebuttal.
>
> ### W2
> Thank you for your comment on Theorem 3.9. We believe there may have been a lack of clarity from our part about its purpose and formulation. The theorem compares the performance of the primary objective between our bilevel optimization approach and the Lagrangian approach, not directly addressing the computation of Demographic Parity (DP). It involves two loss functions (See Fig. 3 in attached PDF): one for the main objective, such as accuracy, and another for the fairness constraint, which may be DP but is not limited to it. The theorem concludes that our bilevel approach can achieve better performance on the primary task under certain conditions compared to the Lagrangian method, while still providing Pareto solutions near local minima.
>
> ### W3
> We appreciate your observation about the connection between our theoretical framework and practical implementation. In response, we have added a section comparing bilevel optimization and Lagrangian regularization, presenting empirical results that validate our theoretical claims and show where our approach excels in balancing accuracy and fairness. We’ve also included a section on how our theoretical assumptions apply in real-world scenarios. While our method yields Pareto optimal solutions under certain conditions, we acknowledge that other methods can also produce different Pareto optimal solutions, and we have discussed the scope of our theoretical guarantees in relation to the broader field of fairness-aware machine learning.
>
> ### W4
> We modified the related work section to position our work better in the literature.
>
> ### W5
> We have addressed several key points in our paper. First, we added a section on practical implementation, discussing hyperparameter selection and its role in providing finer control over balancing accuracy and fairness, along with guidelines for choosing these parameters (See a glimpse of it in the attached PDF, Fig. 1 and 2). Second, we clarified that our method can be applied as a downstream task on pre-trained models, allowing for fairness enhancements without full retraining, as demonstrated with ResNet on CelebA. Third, we discussed extending our method to handle other fairness constraints, with theoretical guidance provided in the appendix. Additionally, we included limitations of our work in global rebuttal.
>
> ## Comments
> The Related Work section was thoroughly revised, categorizing classic multi-objective optimization methods into two groups: common regularization-based methods, including adversarial debiasing, fair representation learning, and Lagrangian optimization, and less common approaches, such as gradient-based methods and transfer and meta-learning approaches. Two mentioned papers were added to the latter group and discussed. The advantages of bilevel optimization were summarized at the end of the section.
>
> ## Questions
> ### Q1
> We provide a theoretical analysis of the neighborhood around local minima in the loss landscape. Due to the convexity of the loss in this area, a sufficiently small step size can lead to strict improvements in the loss. For more details, please refer to item 1 in the assumption discussion in the global rebuttal.
>
> ### Q2
> In practical scenarios, the learning rate is a tunable hyperparameter specific to the task, and the Lipschitz constant is not required for implementation.
>
> ### Q3
> Most modern deep learning architectures satisfy this assumption, as it is met when the network has the capability and parameters to fit all the data. For more details, please refer to item 3 in the assumption discussion in the global rebuttal.
>
> ### Q4
> We apologize for the notation error in the DP definition on Line 212, where  $\theta_1$  and  $\theta_2$  should represent the same set of parameters for both  $a=0$  and  $a=1$ . This will be corrected in the revised paper. Regarding the proof of Lipschitz continuity for DP, the proof is correctly formulated, focusing on the difference between DP loss calculated with two parameter sets. Here,  $f_1$  and  $f_2$  refer to the same function evaluated with different parameters, consistent with DP computation practices \[20,on manuscript\]. The Lipschitz continuity being proven relates to parameter changes, showing that small parameter adjustments lead to bounded changes in the DP metric. The proof establishes that if the underlying function is Lipschitz continuous with respect to its parameters, then the DP metric is also Lipschitz continuous concerning those parameters.
>
> ### Q5
> Please refer to the global rebuttal. We will include a comprehensive discussion in the revised paper.
>
> ### Q6
> Our method’s performance varies due to differences in data handling and augmentation techniques used by baselines, which can affect the loss landscape. While it is not universally superior, our approach consistently demonstrates competitive performance across diverse scenarios, underscoring the complex nature of fairness-accuracy trade-offs in machine learning.
>
> ## Minors
> Thank you for your meticulous attention to detail. All minor issues have been corrected.
>
> ## Limitations
> We have added Limitations and future works to the global rebuttal and manuscript.
> ### L1
> Please refer to "Assumption Discussion" in global rebuttal.
> ### L2
> Thank you for your comment! We have added several ablation studies to the paper. Figures 1 and 2 in attached PDF provide a glimpse of two of these studies.
> ### L3
> Yes, Softmax is a key activation function that is not Lipschitz continuous, while ELU and ReLU are. We have also included a theoretical proof of Lipschitz continuity for another common fairness criterion, equalized odds.

---

> > ### Comment · Reviewer_dbnD · 2024-08-08
> > **Acknowledging response**
> >
> > I thank the authors for their careful rebuttal and revisions of the manuscript.
> >
> > While I believe that the manuscript would benefit from addressing some of the limitations which are currently mentioned as "future work", I agree that this is a novel and interesting take on fairness and the paper is sound and includes a broad array of experiments. Therefore, I have increased my score.

---

### Official Review · Reviewer_UWLB · 2024-07-10

**Soundness:** 3
**Presentation:** 3
**Contribution:** 3
**Rating:** 6
**Confidence:** 4

**Summary:**

The paper proposes and justifies a bi-level optimization approach to optimizing empirical risk minimization objectives when additional fairness constraints need to be considered. Using assumptions of Lipschitz continuity and local convexity, they prove that a bi-criteria (accuracy + fairness) problem is equivalent to bi-level optimization. Key to their approach is to having separate parameters for fairness and accuracy. Practically, accuracy and fairness layers are made for neural networks. Extensive experiments are done.

**Strengths:**

- The paper provides a nice motivation for considering bi-level optimization over the typical fairness regularization approaches.
- Theorem 3.9 is nice to show that the Largrangian approach (under certain conditions) is an upper bound of the bi-level approach.
- Experiments seem promising and extensive.

**Weaknesses:**

- Missing experimental details. Compute time of baseline and approaches not presented. Neither is the number of parameters used in each approach. Additionally, some approaches / baselines in the appendix aren't well specified, eg, FairGAT and FairGCN.
- The convexity assumption does restrict the theory.
- Two separate sets of parameters are required. This limits Theorem 3.9's relevance to the typical fairness regularized cases.

**Questions:**

1. What is the compute time of approach? Does the bi-level optimization approach take longer? How do the number of parameters compare? Basically, I am wondering if due to the separate set of parameters needed in the bi-level optimization, the proposed approaches architecture has a larger number of parameters than the baselines. (and if so, would increasing parameter sizes of other approaches increase fairness)
2. What does the overparameterization condition in Theorem 3.9 mean? I am a bit confused about why its needed in Line 186 / 187.

**Limitations:**

Adequate.

---

> ### Author Rebuttal · Authors · 2024-08-06
>
> Thank you for your thoughtful feedback. We have addressed your questions and concerns below. If you have any further questions, we would be glad to discuss them.
>
> ## Weaknesses
> ### W1
> We acknowledge that some important information was missing from our initial presentation. To address these concerns, we have made the following additions and clarifications:
>
> 1. Compute time and parameter comparison: We have now included a detailed comparison of compute time with the same number of parameters between our method and the Lagrangian approach in the global rebuttal. This comparison is particularly relevant as the Lagrangian method serves as our natural baseline, given the theoretical focus of our work.
>
> 2. Focus on Lagrangian comparison: We want to emphasize that our primary goal was to demonstrate the theoretical results in practice. As such, we focused our direct comparisons on the Lagrangian method, which is most closely related to our approach in terms of problem formulation. We have now made this focus clearer in our experimental setup section.
>
> 3. Other baselines: While we included results from other approaches for context, we acknowledge that a direct comparison of computational complexity or parameter count with these methods may not be fair or meaningful. These methods approach the fairness problem in fundamentally different ways, making such comparisons non-trivial. We have added a note in our experimental section to clarify this point.
>
> 4. Specification of approaches in the appendix: We have added proper citations and brief descriptions of all methods mentioned in the appendix to provide better context. This includes FairGAT, FairGCN, and others.
>
> We appreciate the reviewer’s feedback, which helped improve the transparency of our experimental setup and the validity of our comparisons, particularly concerning the Lagrangian method.
>
> ### W2
> We'd like to clarify that our theory doesn't assume convexity of the entire neural network optimization problem, but rather convexity near local optima. We have added a new section to the paper discussing how our assumptions translate to real-world scenarios, which addresses this point in detail.
> In this new section, we explain that while neural network loss landscapes are generally non-convex, recent research [1] suggests they often exhibit locally convex regions around minima, especially in overparameterized networks. Our assumption of convexity near local optima aligns with this understanding.
> In practice, this assumption translates to the behavior of neural networks as they converge during training. As long as the network converges, it will likely encounter these locally convex regions along its optimization path. Our theory applies particularly well in these parts of the optimization process.
> As networks approach convergence, our assumption of local convexity becomes increasingly valid, making it applicable to well-designed models trained on suitable datasets. Practitioners can rely on this aspect of our theory if their models show convergence. The clarification and new section in our paper highlight how our theoretical framework aligns with practical neural network optimization. By focusing on local rather than global convexity, we significantly broaden the applicability of our theory to real-world scenarios.
>
> ### W3
> We appreciate the reviewer's insightful observation regarding the separate sets of parameters and their impact on Theorem 3.9's relevance to typical fairness regularized cases. We'd like to clarify that the number of parameters in both our bilevel approach and the regularization method are the same, and both methods optimize parameters within the network (please see Fig. 3 in attached PDF). This allows for a meaningful theoretical comparison between the approaches.
> In practice, the separation of parameters into accuracy ($θ_a$) and fairness ($θ_f$) sets in our approach allows for more fine-grained control over the optimization process. This can potentially lead to finding better solutions than single-objective regularization methods, as demonstrated in our experimental results.
>
> ## Questions
> ### Q1
> To address concerns about experimental details, we have enhanced our paper in two key ways. First, we added a new section that directly compares our method to the Lagrangian approach, demonstrating that our method outperforms it with the same number of parameters. Second, we included a comprehensive analysis that examines the computational complexity of both bilevel and regularization training processes (summary in global rebuttal), along with empirical timing results. These additions provide a clear picture of our method’s efficiency and performance relative to traditional regularization techniques.
>
> ### Q2
> The overparameterization condition refers to a scenario where the model has more parameters than necessary to fit the training data perfectly. In the context of our theorem, specifically in lines 186-187, this condition is needed for the following reasons:
>
> 1. Flexibility: Overparameterization provides the model with additional flexibility to find solutions that satisfy both accuracy and fairness objectives simultaneously.
>
> 2. Existence of Multiple Optima: It ensures the existence of multiple parameter configurations that can achieve optimal performance on the primary (accuracy) objective. This is crucial for our bilevel optimization approach, as it allows the model to choose among these configurations to optimize the secondary (fairness) objective.
>
> 3. Theoretical Guarantees: The condition allows us to establish theoretical guarantees about the performance of our bilevel approach compared to the Lagrangian method.
>
> #### References:
> 1. Zeyuan Allen-Zhu, Yuanzhi Li, and Yingyu Liang. Learning and generalization in overparameterized neural networks, going beyond two layers. Advances in neural information processing systems, 32, 2019.

---

> > ### Comment · Reviewer_UWLB · 2024-08-12
> >
> > I thank the authors for providing a thorough response. The additional clarification of W1, Q1, and Q2's response sufficiently answers my question and will bring further clarity on the empirical components of the paper.
> >
> > ### Follow up W3
> > The clarification on W3 was also helpful alongside the figure provided in the extra pdf. It is an interesting observation that the distinct separation of parameters was shown to be superior in settings. One follow up question regarding this:
> >
> > **Q:** Do the primary and secondary parameters sets need to be disjoint?
> >
> > I though it was mentioned somewhere, but I couldn't find it when revisiting the paper. It would be useful to state the answer to this question when initial introducing Eq. (1).
> >
> > This information would be useful in architecture design given the generality of the current results. Although not necessary for this submission, more complicated architectures (eg, siamese-network, SimCLR) beyond the layer position ablation in the pdf would be an interesting follow up work.
> >
> > ### Comment on W2
> >
> > Just wanted to state that I agree with the response of the authors on this. I was merely stating it as a shared weakness of bilevel optimisation / non-convex optimization. But I don't think it should harm the acceptability of the paper.

---

> > > ### Author Response · Authors · 2024-08-12
> > >
> > > Thank you for your follow-up question and the opportunity to clarify this important aspect of our method.
> > > Regarding your question about the primary and secondary parameter sets:
> > > In our implementation, the parameter sets are indeed disjoint. There is no overlap between the two sets of parameters. The gradients of the two objective functions (primary and secondary) are calculated for the whole network, but the two sets of parameters are optimized separately.
> > > We will add an explicit statement when introducing Equation (1) to specify that $\theta_p$ and $\theta_s$ are disjoint sets.
> > > We agree that exploring our method with more complex architectures like siamese networks or SimCLR would be an interesting direction for future work. While it's beyond the scope of the current paper, we'll add this suggestion to our future work section, highlighting how the disjoint parameter structure might be adapted or reconsidered for these more complex architectures.
> > > We appreciate your thoughtful review and suggestions. Your feedback has been invaluable in helping us improve the clarity of our work.

---

### Official Review · Reviewer_QoV4 · 2024-07-14

**Soundness:** 2
**Presentation:** 3
**Contribution:** 2
**Rating:** 5
**Confidence:** 3

**Summary:**

A common approach to bias mitigating in machine learning is to add a regularization term to the loss function that penalizes deviation from fairness. This is what the current paper calls the Lagrangian approach. It has long been know that this is not an effective approach to multi-objective optimization. The paper proposes to use bilevel optimization to tackle this challenge.

**Strengths:**

The paper addresses an important problem in machine learning fairness. Rather than devising a novel fairness metric as is often the focus in this field, this paper studies a more fundamental problem: how do we get solutions on the Pareto front?

The technique employed, bilevel optimization, has not been previously applied in this context, to the best of my knowledge. (However, I did not get a good understanding of the history of bilevel optimization from this paper.)

The method, as presented in Algorithm 1, is alluringly simple.

**Weaknesses:**

The following identified weaknesses could be due to my own misunderstanding:

*I could not understand how one would decide on which are accuracy parameters and which are fairness parameters in a neural network architecture. I could not find guidance on this in the paper.

*Except for the simplest types of neural networks, I can't see how a general neural network can satisfy Assumption 3.4

*Where does bilevel optimization sit in the larger context of multi-objective optimization techniques? I think this needs to be explicitly spelled out.

**Questions:**

*Line 51: "we introduce a novel method that can be trained on existing datasets without requiring any alterations". Is this an accurate characterization? There certainly is a modification to the training algorithm?
*"Leader-follower" terminology: is this standard in bilevel optimization? Can you add some references?
*In assumption 3.3, $\hat \theta_s$ appears to be undefined
*Could you give some example of common activation functions that are Lipschitz continuous? This would help the reader understand how realistic Lemma 3.5 is for common activation functions.
*Line 208: "demographic loss function of given layers..." what do you mean by "of given layers"?
*Is there an equivalent of Theorem 3.11 for other fairness metrics considered in your paper? Like equal opportunity? or Equal odds?
*Line 219: "in practice, the model $f$ can be implemented as a neural network with separate layers for accuracy and fairness". This is one of my main confusions about the paper. How does one decide which layers are accuracy layers and which ones are fairness layers? Some of your experiments concern NNs with just one hidden layer, so then which are accuracy and fairness layers there?
*Line 298: "The theoretical analysis, particularly Theorem 4.6, establishes the properties of the optimal solution under certain assumptions, which are not limited to specific datasets or network architectures." I could not find Theorem 4.6. I also think your theoretical analysis is very limited due to the lipschitz continuity assumption. Most NNs used in practice woudl violate this assumption?

**Limitations:**

I would appreciate a further discussion on the limitations of the theoretical analysis. In particular, I believe the theory is highly limited in terms of the type of NNs it can be applied to.

---

> ### Author Rebuttal · Authors · 2024-08-06
>
> Thank you for your thoughtful feedback. We have addressed your questions and concerns below. If you have any further questions, we would be glad to discuss them.
>
> ## Weaknesses
> ### W1
> In our approach, the separation of accuracy and fairness parameters is a key design choice that allows for the bilevel optimization. Typically, accuracy parameters ($θ_a$) are those in the main network layers directly contributing to task performance, while fairness parameters ($θ_f$) are in additional layers specifically designed to address fairness concerns.
> To address this lack of clarity, we have added a new figure to the paper that visually illustrates how accuracy and fairness parameters are typically separated in different networks (Attached PDF, Fig. 3).
> For instance, in a simple feedforward network for binary classification, the structure might be:
>
> Input -> Main Layers ($θ_a$) -> Fairness Layers ($θ_f$) -> Output Layer ($θ_a$)
>
> The architecture of our approach can vary based on the specific task and dataset, and the separation of accuracy and fairness parameters may not always be straightforward. Practitioners may need to experiment with different configurations to find the most effective split for their use case. This flexibility is a strength of our method, allowing adaptation to various network designs. To enhance clarity, we added an ablation study examining the impact of fairness layer positioning within the network architecture (Attached PDF, Fig. 2), offering insights into how design choices affect model performance and fairness. We appreciate the reviewer’s feedback, which improved the clarity and applicability of our work.
> ### W2
> We appreciate the reviewer’s insightful observation regarding Assumption 3.4 and its applicability to general neural networks. While this assumption may initially seem restrictive, we have added a section to the paper explaining how it can be met in practice. Specifically, Assumption 3.4 can be satisfied through common techniques such as:
> * Bounded Activation Functions: Many popular activation functions, such as sigmoid, tanh, and ReLU (when combined with proper weight initialization), naturally bound the output of each layer.
> * Regularization methods
> * Normalization techniques
>
> These approaches are widely used in modern neural networks, ensuring that many architectures inherently satisfy or can easily be modified to meet this assumption without significantly altering their structure or performance.
> ### W3
> Thank you for your suggestion! We have modified the related work to place our work better in the existing literature.
>
> ## Questions
> ### Q1
> You are correct that our original statement could have been more precise. We have edited it to ".. without requiring any alterations to the data itself (data augmentation, perturbation, etc)."
>
> Unlike some fairness approaches that require data augmentation or perturbation, our method operates on the original datasets without alterations. This allows for direct application to existing datasets without preprocessing, avoiding additional biases or changes to the data distribution. Our approach focuses on modifying the learning process, not the data, which is a key strength of our method.
> ### Q2
> We have added relevant papers and refined the related work section accordingly.
> ### Q3
> $\hat{\theta}_s$ represents the updated values of the secondary objective's parameters.
> ### Q4
> Great insight! We have added a discussion and examples of different activation functions and layers in the appendix. In summary, Linear, Sigmoid, Tanh, ReLU, Leaky ReLU, ELU, and Softplus are Lipschitz continuous, while Softmax, Hard Tanh, and Hard Sigmoid are not.
> ### Q5
> The revised statement now reads:
> "We showed that the demographic parity loss function, when applied to the output of neural network layers, is also Lipschitz continuous (Theorem 3.10)"
> ### Q6
> We know:
> \begin{align}
> \text{EO Difference} = \max(\text{TPR Difference}, \text{FPR Difference})
> \end{align}
>
> Lipschitz constants for true positive rates (TPR) and false positive rates (FPR) can be calculated in the same way as in Theorem 3.10. Therefore, Equalized Odds (EO) possesses a Lipschitz constant, indicating that it is a smooth function.
> ### Q7
> We covered this. (Response to W1)
> ### Q8
> We apologize for the confusion caused by the incorrect theorem number. There is no Theorem 4.6 in our paper; the correct reference should be to Theorem 3.8. This error has been corrected in the text.
>
> We acknowledge your concern about the limitations of the Lipschitz continuity assumption, as many practical neural networks might appear to violate it. To address this, we have added a section discussing how our theoretical assumptions apply to real-world scenarios, highlighting how neural network design and training practices can satisfy or approximate these assumptions. Additionally, we have included an appendix section exploring the Lipschitz properties of various layers, demonstrating that many commonly used types are Lipschitz continuous or can be modified to be so.
>
> Although the Lipschitz continuity assumption may seem restrictive, our analysis shows that it is more applicable to practical neural networks than it initially appears. Many modern architectures, such as CNNs[1], GNNs[2], and GATs[3], either inherently satisfy this property or can be easily adjusted to do so.
>
>
> ## Limitations
> ### L1
> We have added limitations and future works to global rebuttal and revised manuscript.
> #### References:
> 1. Dongmian Zou, Radu Balan, and Maneesh Singh. On lipschitz bounds of general convolutional neural networks. IEEE Transactions on Information Theory, 66(3):1738–1759, 2019.
> 2. Simona Ioana Juvina, Ana Antonia Neacs, u, Jérôme Rony, Jean-Christophe Pesquet, Corneliu Burileanu, and Ismail Ben Ayed. Training graph neural networks subject to a tight lipschitz constraint. Transactions on Machine Learning Research.
> 3. Reference [2] in global rebuttal.

---

> > ### Comment · Reviewer_QoV4 · 2024-08-13
> > **thank you for the detailed response**
> >
> > Thank you for the detailed response. I'm afraid I still find it quite unnatural and confounding to define certain parameters "accuracy" parameters and certain parameters "fairness" parameters. I do not find Figure 3 in the one-page PDF illuminating. It seems completely arbitrary, rather.
> >
> > I would also advise that the authors clearly state that they are not modifying existing architectures. Both Reviewer UWLB (question 1) and the ethical reviewer had the impression that the proposed methodology appends new layers/parameters to existing networks.

---

> > > ### Author Response · Authors · 2024-08-13
> > >
> > > Thank you for your thoughtful feedback. The selection of accuracy and fairness parameters is indeed treated as a hyperparameter choice, which might appear arbitrary. To address this, we have conducted various ablation studies specifically focused on this matter.
> > > The key point is that we need two disjoint sets of parameters for the two objectives. The assignment of network parts to these sets is a hyperparameter decision.
> > >
> > > We appreciate your advice and will clarify the architecture discussion as you suggested.

---

### Official Review · Reviewer_J3Wm · 2024-07-18

**Soundness:** 3
**Presentation:** 3
**Contribution:** 3
**Rating:** 6
**Confidence:** 4

**Summary:**

This paper proposes a novel bilevel optimization framework called FairBiNN for addressing bias and fairness issues in machine learning models while maintaining accuracy. The approach formulates the problem as a Stackelberg game between accuracy and fairness objectives, proving that it yields Pareto-optimal solutions under certain assumptions. Theoretical analysis shows the method performs at least as well as Lagrangian approaches. Experiments on tabular, graph, and vision datasets demonstrate competitive or superior performance compared to state-of-the-art fairness methods in balancing accuracy and fairness metrics like demographic parity and equality of opportunity.

**Strengths:**

1. Proposes a novel bilevel optimization framework with theoretical guarantees for fairness-aware machine learning.
2. Provides rigorous theoretical analysis and proofs for the proposed method.
3. Demonstrates strong empirical results across multiple domains (tabular, graph, vision) and datasets as shown in both main text and appendix.
4. Compares against numerous state-of-the-art baselines.
5. Conducts thorough ablation studies to analyze different components.
6. Provides clear implementation details and hyperparameters for reproducibility.
7. Visualizes results effectively through plots and t-SNE visualizations.

**Weaknesses:**

1. The bilevel optimization framework may be more complex to implement and understand compared to traditional methods.
2. The theoretical guarantees rely on several assumptions that may not hold in all practical scenarios.

**Questions:**

1. How sensitive is the method to the choice of hyperparameters, especially $\eta$?
2. How does the computational complexity compare to the baseline methods?
3. Are there any scenarios where the method might not perform as well as traditional approaches?

**Limitations:**

1. The paper does not explicitly discuss limitations of the proposed approach.
2. Experiments are limited to a few datasets per domain; more diverse datasets could strengthen generalizability claims.
3. The approach is not tested on very large-scale datasets or models

---

> ### Author Rebuttal · Authors · 2024-08-06
>
> Thank you for your thoughtful feedback. We have addressed your questions and concerns below. If you have any further questions, we would be glad to discuss them.
>
> ## Weaknesses
> ### W1
> While our approach may initially appear more complex than traditional methods, we believe its benefits significantly outweigh this drawback. Our experiments demonstrate that the proposed method offers greater training stability compared to common adversarial training techniques in fairness research. The optimization can be implemented through a customized training loop with separate optimizers for accuracy and fairness, which is straightforward in modern deep learning frameworks.
>
> The bilevel framework enhances interpretability by clearly separating accuracy and fairness objectives. This separation allows for more fine-grained control over parameters compared to fairness regularization methods, enabling precise tuning of the trade-off between objectives. This level of control is particularly valuable in sensitive applications where balancing fairness and accuracy is crucial.
> ### W2
> We appreciate your insightful comment regarding the assumptions underlying our theoretical guarantees. We have addressed assumptions' discussion in global rebuttal.
>
> ## Questions
> ### Q1
> We have included ablation study in attached PDF (Fig. 1) showing the impact of varying the parameter $\eta$ on the FairBiNN model. The results revealed a trade-off: increasing $\eta$ enhanced fairness but decreased accuracy, with diminishing returns at higher values. This underscores the importance of tuning $\eta$ to achieve an optimal balance between accuracy and fairness, tailored to the specific needs of each application.
>
> ### Q2
> In our paper, we focused our theoretical analysis and direct comparison on the Lagrangian regularization method, as it serves as a natural baseline for our bilevel optimization approach.
> We'd like to emphasize that the computational complexity of our approach is equivalent to that of the Lagrangian method. Both methods use the same number of parameters and require similar computational resources per iteration. This equivalence in complexity allows for a fair and direct comparison between the two approaches.
> We have included a detailed theoretical analysis of the computational aspects for both our method and Lagrangian regularization in global rebuttal. Additionally, we report practical execution times for both approaches in our experimental results. These comparisons provide a clear picture of how our method compares to Lagrangian regularization in terms of computational efficiency.
>
> Direct complexity comparisons with other baseline methods are challenging due to their varying approaches to ensuring fairness. By focusing on Lagrangian regularization, we provide a clear comparison of our method’s computational efficiency relative to a similar approach. This comparison highlights the practical benefits of our bilevel optimization framework while maintaining comparable computational efficiency.
>
> ### Q3
> While our approach has shown strong performance across various scenarios, there are indeed cases where it might face challenges or not perform as well as traditional approaches.
> One notable scenario is multiclass classification, particularly when using softmax activation in the output layer. The softmax function is not Lipschitz continuous, which is one of the key assumptions in our theoretical framework. This lack of Lipschitz continuity could potentially lead to instability in the optimization process or reduced performance compared to traditional approaches in multiclass settings. Also, in non-stationary environments, Our current framework assumes a static dataset. In scenarios where data distributions change over time, the method might require adaptations to maintain its performance.
>
> ## Limitations
> ### L1
> We have added the limitations and future work section in global rebuttal.
> ### L2 and L3
> We appreciate the reviewer's observation regarding the number of datasets used in our experiments. We acknowledge that testing on a wider variety of datasets could indeed strengthen generalizability claims. However, our dataset selection was deliberate and constrained by several factors in the field of fairness research.
>
> We chose widely used benchmark datasets that are well-established in fairness tasks, such as UCI Adult and Heritage Health for tabular data, Pokec and NBA for graph data, and CelebA for vision. These datasets allow for direct comparison with existing methods. It's important to note that the number of high-quality benchmark datasets specifically designed for fairness tasks is somewhat limited in the field, posing a challenge for all researchers in this area.
>
> As our primary contribution is a theoretical framework, our main goal was to demonstrate that our theoretical results hold in practice. The selected datasets, covering a range of domains and data types, serve this purpose well. While we agree that more datasets could potentially strengthen generalizability claims, we believe that the diversity of domains we covered provides strong evidence for the broad applicability of our method.
>
> We are actively seeking to apply our method to additional datasets as they become available as part of our ongoing research. We believe our current results, spanning multiple domains and dataset types, provide a solid foundation for the practical utility of our theoretical framework. However, we welcome and encourage further testing and application of our method on diverse datasets by the research community to further validate its generalizability.
>
> ### Flag For Ethics:
> All datasets used in this study are publicly available, and no human subjects were involved in the research.

---

> > ### Comment · Reviewer_J3Wm · 2024-08-12
> > **Keep Score**
> >
> > I thank the authors' response. I think the current response does not change too much of my original impression. I will thus keep my score.

---

### Author Rebuttal · Authors · 2024-08-06

# Global Rebuttal

We sincerely thank the reviewers for their constructive comments, which have significantly contributed to the improvement of our work. We have addressed the reviewers’ concerns regarding our method’s assumptions, limitations, and ablation studies.

## Assumption Discussion
We appreciate the reviewer’s insightful comments on the assumptions behind our theoretical guarantees. We have added a detailed discussion in the paper to address these concerns and relate these assumptions to practical scenarios. Here is the summary of it:

1. Convexity Near Local Optima: While neural networks are generally non-convex, research [1] suggests they exhibit locally convex regions around minima, especially in overparameterized networks. Our theory is particularly applicable as the network approaches convergence, which is common in well-designed models on suitable datasets. (Assumption 3.2)

2. Small Steps for Secondary Parameters: The assumption $|θ_s − \hat{θ}_s| ≤ ε$, where ε is sufficiently small, can be met by choosing an appropriate learning rate. This can be achieved through hyperparameter tuning and grid search, which are standard practices in machine learning.(Assumption 3.3)

2. Overparameterization: This assumption aligns well with modern deep learning trends, where models often have more parameters than training samples. As long as the model can overfit the training data, this condition is met. However, we acknowledge that in resource-constrained environments or with extremely large datasets, this might not always be feasible.

3. Lipschitz Continuity: We have added a rigorous analysis of various layers and activation functions in terms of their Lipschitz properties, serving as a guide for practitioners. Common choices like ReLU activations and standard loss functions (e.g., cross-entropy) satisfy this assumption, making it broadly applicable. (Assumption A.7, A.8)

4. Bounded Output: The assumption of bounded output for each layer can be achieved in practice through the use of bounded activation functions and weight regularization methods, which are common in neural network optimization. (Assumption 3.4)

While these assumptions may not apply universally, they are common in many real-world deep learning applications. Our framework provides valuable insights even when assumptions are only approximately met. Strong empirical results across various datasets support our approach’s practical utility.

We hope this discussion clarifies the applicability of our theoretical guarantees and addresses the reviewer's concerns.

## Limitations and Future Work
While our results are promising, our approach has several limitations. The widely used softmax activation function is not Lipschitz continuous, limiting our method’s application to multiclass classification. Future work could explore alternative activation functions that maintain Lipschitz continuity for multiclass problems.

Additionally, attention mechanisms in modern models are not Lipschitz continuous, posing challenges for extending our method to architectures that rely on attention. Research into enforcing Lipschitz continuity in attention layers, such as using LipschitzNorm [2], a simple and parameter-free normalization technique applied to attention scores, shows potential for enhancing performance in deep attention models, and integrating this with our framework.

Our theoretical analysis mainly provides guarantees in comparison to regularization methods, offering improvements in fairness but not absolute fairness guarantees. Expanding the framework to include direct fairness guarantees would increase its applicability.

We did not validate the method with dataset augmentation, a common practice to improve generalization. Future work should assess how our method interacts with data augmentation and its impact on fairness properties.

Currently, our implementation focuses on demographic parity, but real-world applications often require multiple fairness metrics. Extending our method to address multiple fairness constraints would make it more versatile.

Addressing these limitations presents opportunities for future research to enhance the applicability and effectiveness of fair machine learning methods in diverse scenarios and architectures.

## Computational Complexity Analysis
Let's define the following variables:

- $n$: number of parameters in $\theta_p$
- $m$: number of parameters in $\theta_s$
- $C_f$: cost of computing $f$ and its gradients
- $C_\phi$: cost of computing $\phi$ and its gradients

;where $\theta_p$ and $f$ are related to primary objective and $\theta_s$ and $\phi$ are related to secondary objective.

Based on the Lagrangian and Bilevel update rules, computational complexity per iteration for both approaches: $O(C_f + C_\phi + n + m)$

### Empirical Comparison:

While theoretical complexity analysis indicates similar costs for both methods, we conducted empirical tests to compare their runtime performance on the Adult and Health datasets using FairBiNN and Lagrangian methods after 10 epochs of warmup. These experiments, conducted on an M1 Pro CPU, showed no significant difference in average epoch time between the two methods, supporting our theoretical analysis.

#### Table 1: Average epoch time (in seconds) for FairBiNN and Lagrangian methods

| Method     | Adult Dataset (s) | Health Dataset (s) |
|------------|-------------------|--------------------|
| FairBiNN   | 0.62              | 1.03               |
| Lagrangian | 0.60              | 1.05               |



#### References:
1. Zeyuan Allen-Zhu, Yuanzhi Li, and Yingyu Liang. Learning and generalization in overparameterized neural networks, going beyond two layers. Advances in neural information processing systems, 32, 2019.
2. George Dasoulas, Kevin Scaman, and Aladin Virmaux. Lipschitz normalization for self-attention layers with application to graph neural networks. In International Conference on Machine Learning, pages 2456–2466. PMLR, 2021.

---

### Decision · Program_Chairs · 2024-09-25

**Decision:**

Accept (poster)

**Comment:**

The current paper has its strengths in its theoretical contributions towards introducing a novel bilevel optimization framework to address the fundamental problem of finding solutions on the Pareto front in ML fairness. The theoretical rigor, including guarantees and proofs, are significant contributions and seems to be unanimously agreed upon by the reviewers. The ablation studies presented in the paper, seems to highlight the practical potential of the proposed approach. The broad array of experiments demonstrates the method's efficacy across various domains as well as stability of their approach.

There were concerns raised by the reviewers on the practicality of theoretical assumptions with respect to real-world scenarios as well as lack of discussions towards limitations of the proposed approach. Going over the rebuttal as well as the discussions between the authors and reviewers, I see that the authors seem to have addressed this in the context of current prevalent practices in deep learning implementations. Including them in the revision should bring more clarity over practical utility of the paper. Authors seem to have also provided additional discussion on choice of baselines and comparison with Lagrangian regularization method which seems fair considering their theoretical results. The reviews and rebuttal also highlights that a discussion over limitations of the proposed approach needs to be included in the final version.

Considering the paper's contributions novel and significant, I **recommend acceptance** with revisions addressing the identified concerns.

Following are the specific items that the authors should address -

1. **Limitations**: Include a brief discussion in the main text and a detailed appendix section outlining the limitations of the current approach.
2. **Ethics**: Add an ethics section to discuss potential ethical implications and considerations related to the proposed method.
3. **Parameter Selection**: Include a figure in main text clarifying parameter selection in neural networks. Explicitly stating correspondence of parameters to notations in theory.
4. **Assumptions**: Add a detailed section in Appendix which clarifies how different components in deep networks fit or do not fit under the proposed assumptions.
5. Computational Complexity Analysis to be added somewhere.
6. Update the discussion in the experiments section explaining the choices of baselines for comparison with the Lagrangian approach.
7. **Practical Implementation**: Address practical implementation concerns like hyperparameter selection, model reusability, and extension to multiple fairness constraints.

Addressing these concerns and incorporating the suggested revisions will significantly strengthen the paper and enhance its impact.